# EZ-HOI: VLM Adaptation via Guided Prompt Learning for Zero-Shot HOI Detection

Qinqian Lei[1]    Bo Wang[2]    Robby T. Tan[1,3]
[1]National University of Singapore
[2]University of Mississippi
[3]ASUS Intelligent Cloud Services (AICS)
`qinqian.lei@u.nus.edu`, `hawk.rsrch@gmail.com`, `robby_tan@asus.com`

## Abstract

Detecting Human-Object Interactions (HOI) in zero-shot settings, where models must handle unseen classes, poses significant challenges. Existing methods that rely on aligning visual encoders with large Vision-Language Models (VLMs) to tap into the extensive knowledge of VLMs, require large, computationally expensive models and encounter training difficulties. Adapting VLMs with prompt learning offers an alternative to direct alignment. However, fine-tuning on task-specific datasets often leads to overfitting to seen classes and suboptimal performance on unseen classes, due to the absence of unseen class labels. To address these challenges, we introduce a novel prompt learning-based framework for Efficient Zero-Shot HOI detection (EZ-HOI). First, we introduce Large Language Model (LLM) and VLM guidance for learnable prompts, integrating detailed HOI descriptions and visual semantics to adapt VLMs to HOI tasks. However, because training datasets contain seen-class labels alone, fine-tuning VLMs on such datasets tends to optimize learnable prompts for seen classes instead of unseen ones. Therefore, we design prompt learning for unseen classes using information from related seen classes, with LLMs utilized to highlight the differences between unseen and related seen classes. Quantitative evaluations on benchmark datasets demonstrate that our EZ-HOI achieves state-of-the-art performance across various zero-shot settings with only 10.35% to 33.95% of the trainable parameters compared to existing methods. Code is available at `https://github.com/ChelsieLei/EZ-HOI`.

## 1 Introduction

Human-Object Interaction (HOI) detection localizes human-object pairs and identifies the interactions. HOI has various practical applications, including robot manipulations, human-computer interaction, and human activity understanding [38, 41, 24, 32, 26, 1]. Additionally, it is a building block for related tasks such as action recognition, visual question answering, and image generation [18, 24, 4, 43, 11, 54]. However, the limited generalization ability of many existing HOI detectors makes zero-shot HOI detection particularly challenging, as it requires models to identify unseen HOI classes.

Vision-Language Models (VLMs) [47, 29, 2, 30] have gained popularity due to their extensive knowledge bases and effective ability to process and correlate complex patterns across visual and text data. Consequently, a group of methods has been developed to align HOI visual features with those of VLMs, leveraging the extensive knowledge from these models [51, 33, 44, 41, 4]. This alignment ensures that both the HOI model and the VLM can extract similar features to represent the same concept (e.g., an action). The degree of feature alignment can be quantified using cosine similarity, which measures the similarity between feature vectors. However, aligning with VLMs requires

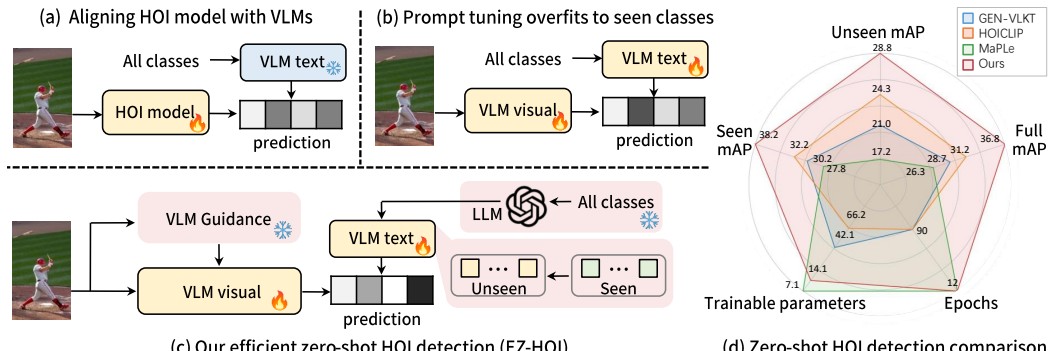

Figure 1: Comparison of zero-shot HOI detection paradigms. (a) Methods that align HOI features with fixed VLMs [44, 33, 4, 41]. (b) Prompt learning methods to adapt VLMs for downstream tasks [22, 57]. (c) Our approach, which adapts VLMs to HOI tasks without compromising VLM generation capabilities. (d) Unseen, seen, and full mAP indicate the performance for unseen-verb, seen-verb, and full sets on the HICO-DET dataset [6]. Our EZ-HOI shows superior performance in these categories, with competitive trainable parameters and training epochs.

training transformer-based models, which is computationally expensive and leads to extended training time and significant difficulties.

An alternative approach involves adapting VLMs for HOI tasks, bypassing the demanding alignment process and leveraging VLM capabilities for action understanding in HOI detection. Prompt tuning, which adapts VLMs to various downstream tasks with a small number of learnable parameters, has shown considerable efficacy [62, 61, 19, 21]. One notable method, MaPLe [22], utilizes multi-modal prompt tuning specifically for image classification tasks. However, since adaptation typically involves only seen class labels, the adapted VLMs often overfit to seen classes but struggle to deal with unseen classes in zero-shot settings [62, 7, 9, 22]. Consequently, this limitation results in suboptimal performance for unseen classes in zero-shot HOI detection.

In this paper, we introduce a novel method, Efficient Zero-Shot HOI detection (EZ-HOI), to enhance VLM adaptation via guided prompt learning with information from foundation models (e.g., LLM and VLM). We design both learnable text and visual prompts, leveraging fixed LLM and VLM to guide the adaptation process. Specifically, the text prompts are designed to capture detailed HOI class insights from an LLM, while the visual prompts are tailored to incorporate external visual semantics from a VLM. However, as the training datasets contain images with seen-class labels alone, the trainable prompts are naturally optimized for these seen classes rather than the unseen ones.

To address this limitation, we develop the Unseen Text Prompt Learning (UTPL) to leverage prompts from related seen classes effectively. We begin by measuring the relationship between HOIs using cosine similarity of text embeddings, which helps identify closely related seen classes for each unseen one. After establishing these connections, we enhance the learning of unseen prompts based on those from selected seen classes. To capture the distinctions between unseen and seen classes, we incorporate an LLM. This LLM provides what we call "disparity information", enhancing the learning of the distinctions and similarities between seen and unseen classes. Additionally, we enhance our approach by introducing intra- and inter-HOI feature fusion techniques following the visual encoder. Our method achieves competitive performance on established benchmarks in various zero-shot settings while requiring significantly fewer trainable parameters.

As illustrated in Fig. 1, we compare our method with two existing paradigms: (a) aligning HOI features to a fixed VLM and (b) adapting a VLM for HOI tasks. Unlike the existing approaches, our method facilitates VLM adaptation to HOI tasks and demonstrates effective generalization to unseen classes. Fig. 1(d) highlights that our method is competitive in terms of performance, model parameter efficiency, and training duration.

In summary, our contributions are as follows:

- We introduce EZ-HOI, a novel framework for zero-shot HOI detection that adapts a VLM to HOI tasks via guided prompt learning with foundation model information, enhancing the adapted VLM's generalizability in zero-shot HOI detection.

- We propose the UTPL module, which extracts information from related seen classes for the unseen learnable prompts. This module mitigates overfitting to seen classes, a common issue in task-specific VLM adaptations, improving performance on unseen classes.

- Our EZ-HOI method achieves state-of-the-art performance in various zero-shot settings while significantly reducing trainable parameters and mitigating training challenges. It lowers trainable model parameters by 66% to 78% compared to existing methods, demonstrating enhanced efficiency and effectiveness in zero-shot HOI detection.

## 2 Related Work

**Human-Object Interaction Detection** Human-Object Interaction (HOI) detection involves identifying human-object pairs and recognizing their interactions, serving as a fundamental component for various downstream tasks in computer vision [53, 6]. HOI detection methods are typically categorized into two classes: one-stage and two-stage. One-stage methods simultaneously generate all outputs, including human bounding boxes, object bounding boxes and categories, and interaction classes. Recent advancements in one-stage detectors leverage transformer architectures, delivering promising performance [8, 46, 48, 63, 55, 52, 24]. An example is HOITrans [63], which utilizes the transformer's encoder and decoder to extract interaction features. The output features are then processed through multi-layer perceptrons to produce all output predictions at once. On the other hand, two-stage approaches divide HOI detection into two tasks: object detection and HOI classification [10, 14, 58]. Since separation allows each module to specialize, two-stage HOI detection is more memory-efficient [59]. Recent developments have seen the integration of transformer-based architectures into two-stage designs, which have shown promising results [59, 60, 45, 27]. Our method also falls into the two-stage design category.

**Zero-Shot HOI Detection** Prior zero-shot HOI detection efforts primarily address unseen composition settings, where models encounter action and object classes separately but not as combinations. To address this, several methods employ compositional learning strategies aimed at tackling the unseen-composition problem [14, 16, 15, 27]. However, compositional learning falls short in addressing unseen verb zero-shot HOI detection scenarios. Given the substantial image understanding capabilities of VLMs, they offer promising potential for enhancing HOI detection in zero-shot settings [2, 29, 30, 47, 37, 36], where labels for unseen HOI classes are absent from the training dataset. Recent studies have explored aligning HOI visual features with VLM [33, 44, 4, 41]. However, this approach requires training transformer-based models, which are computationally expensive. Consequently, aligning HOI models with VLMs is demanding, resulting in extended training times and significant training challenges.

**Prompt Learning** Prompt learning has gained popularity for adapting VLMs to downstream tasks [61, 62, 7, 9, 19, 23, 25, 40]. Context Optimization (CoOp) [62] refines the prompt input of the text encoder by combining learnable domain-shared prompt tokens with class prompt tokens. MaPLe [22] integrates learnable domain-shared text tokens with visual learnable tokens. This combination leverages the highly connected text and visual encoders in VLMs, facilitating the sharing of cross-domain information and benefiting both text and visual domains. However, fine-tuning VLM relies on training datasets with only seen class labels, which causes adapted VLMs to excel with seen classes but encounter difficulties with unseen ones in zero-shot scenarios. Consequently, this limitation results in suboptimal performance for unseen classes in zero-shot HOI detection.

## 3 Methodology

We start with adapting a pre-trained CLIP model [47] to zero-shot HOI tasks using an innovative prompt learning approach. We design two sets of learnable prompts: text and visual. The visual prompts are derived from the text prompts by using projection layers. We denote learnable text prompts as $h_T$ and learnable visual prompts as $h_V$:

$$h_T = \mathrm{Proj}(h_V), \tag{1}$$

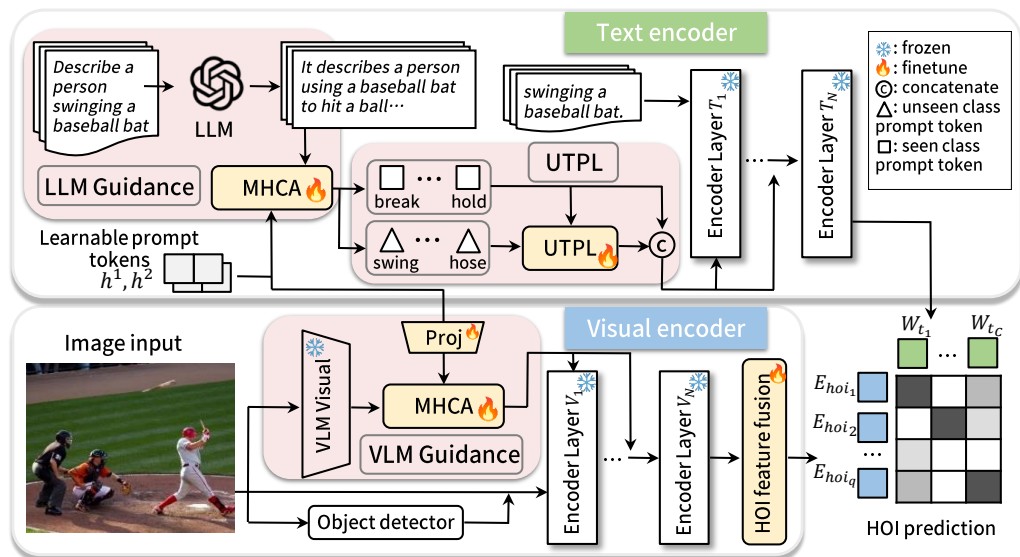

Figure 2: Overview of our EZ-HOI framework. Learnable text prompts capture detailed HOI class information from the LLM. To enhance their generalization ability, we introduce the Unseen Text Prompt Learning (UTPL) module. Meanwhile, visual learnable prompts are guided by a frozen VLM visual encoder. These learnable text and visual prompts are then separately input into the text and visual encoder. Finally, HOI predictions are made by calculating the cosine similarity between the text encoder output and the HOI image features. MHCA denotes multi-head cross-attention.

where $h_T \in \mathcal{R}^{p*d_t}$, and $h_V \in \mathcal{R}^{p*d_v}$. $d_t$ is the dimension of the text feature, and $d_v$ is the dimension of the visual feature. $p$ represents the number of tokens we design for each learnable prompt.

Let $\mathbb{V} = \{v_1, v_2, \cdots, v_{N_v}\}$ as the verb set and $\mathbb{O} = \{o_1, o_2, \cdots, o_{N_o}\}$ as the object set. Then the HOI set include all feasible verb-object pairs $\mathbb{C} = \{\text{hoi}_i = (v_i, o_i)|v_i \in \mathbb{V}; o_i \in \mathbb{O}\}$. We set $C$ as the total number of HOI classes. Since it is impractical for any datasets to include comprehensive HOI classes, researchers propose zero-shot HOI detection settings to encourage the generalization of HOI detectors to unseen classes $\mathbb{C}_{\text{unseen}}$ in inference [33]. Denote the seen HOI class set as $\mathbb{S}$ and the unseen HOI class set as $\mathbb{U} = \{\text{hoi}_i \mid \text{hoi}_i \notin \mathbb{S}, \text{hoi}_i \in \mathbb{C}\}$. Please refer to Appendix 7.9 for detailed definitions of the different zero-shot setting unseen set splits.

## 3.1 LLM and VLM Guidance for Learnable Prompts

Prompt learning techniques for VLMs are characterized by their ability to adapt large, pre-trained models to specific tasks using relatively small amounts of task-specific data. This often leads to a diminished generalization ability of the fine-tuned models for unseen classes [62, 7, 9, 22]. Given that foundation models, such as LLMs and VLMs, have demonstrated substantial knowledge capacity [29, 47], leveraging guidance from these models can improve performance on unseen HOI classes that are absent in the training data.

**Text Prompt Design** As shown in Fig. 2, we design input prompts, *"Describe a person <acting> <object> and focus on the action"* related to each HOI class, for an LLM. The output from LLM contains richer semantic information with specific HOI class descriptions. Please refer to Appendix Section 7.2 for a detailed explanation with examples.

Then, we use a CLIP text encoder to obtain the text embedding for this HOI class description. We process text embeddings for all HOI classes including both seen and unseen in parallel, so the whole text embeddings are denoted as $\mathcal{F}_{\text{txt}} \in \mathcal{R}^{C*d_t}$ and $\mathcal{F}_{\text{txt}} = [f_{\text{txt}_1}, f_{\text{txt}_2}, \cdots, f_{\text{txt}_C}]$. Then, the learnable text prompts $\hat{h_T}$ can be obtained by:

$$\hat{h_T} = W_{\text{up}} \cdot \text{MHCA}(Q = W_{\text{down}} \cdot h_T; \ K, V = W_{\text{down}} \cdot \mathcal{F}_{\text{txt}}) + h_T. \tag{2}$$

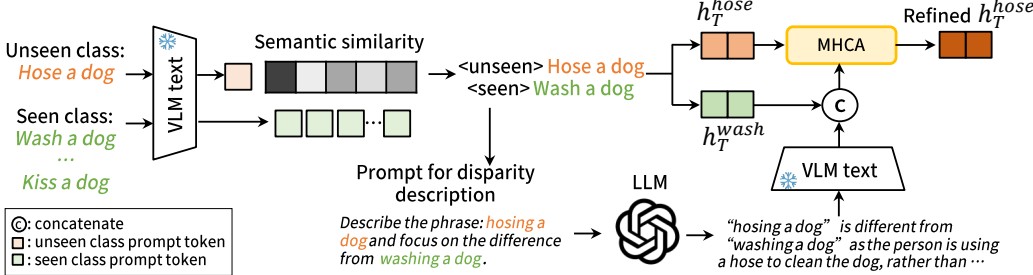

Figure 3: Detailed architecture of Unseen Text Prompt Learning (UTPL). In the figure, we take the "hose a dog" unseen HOI class in the unseen-verb zero-shot setting as an example. We first utilize the HOI class text embeddings to identify the most connected seen HOI class to "hose a dog". After selecting the seen class, we generate an input prompt to obtain disparity information from LLM. Finally, the unseen learnable prompt learns from the selected seen class prompt and the disparity information through MHCA.

where $\hat{h_T} = [\hat{h_{T_1}}, \hat{h_{T_2}}, \cdots, \hat{h_{T_C}}]$, and $\hat{h_{T_i}} \in \mathcal{R}^{p*d_t}$. Each class has its specific learnable text prompts after the process of Eq. 2. $W_{up}$ and $W_{down}$ indicate the up-projection and down-projection layers. MHCA means the multi-head cross attention [50]. In Eq. 2, we only aggregate the useful information from $\mathcal{F}_{txt}$ using learnable attention, as the information provided by LLM may not all be relevant for our task. Moreover, to keep trainable parameters small, we apply a down-projection layer $W_{down}$ before MHCA to reduce the feature dimension, and an up-projection layer $W_{up}$ afterward. The same design strategy is used in Eq. 3 and Eq. 5. Later, the prompts corresponding to unseen classes are further refined, as detailed in Section 3.2.

**Visual Prompt Design** Visual prompt learning in our method is facilitated by a pre-trained Visual Language Model (VLM) visual encoder, specifically the CLIP visual encoder [47]. The pre-trained CLIP model inherently possesses knowledge of unseen HOI classes, offering richer visual semantics compared to models trained solely on task-specific datasets. The CLIP visual feature is denoted as $f_{vis}^{\mathcal{I}}$ for an image $\mathcal{I}$, and we enhance the visual learnable prompts $h_V$ by:

$$\hat{h_V} = W_{up} \cdot \text{MHCA}(Q = W_{down} \cdot h_V; \ K, V = W_{down} \cdot f_{vis}^{\mathcal{I}}) + h_V. \tag{3}$$

Since the frozen VLM visual encoder can extract features for unseen HOIs, $\hat{h_V}$ aggregates information from these visual features, improving performance on unseen HOIs.

## 3.2 Unseen-Class Text Prompt Learning (UTPL)

Since the training data has no labeled image for unseen HOI classes, learnable text prompts for seen classes are inevitably optimized better than unseen classes. Therefore, we refine the unseen-class text prompts by learning from closely related seen-class prompts. Specifically, we denote one unseen HOI class as $u$, and the learnable text prompt tokens as $\hat{h_{T_u}}$, and denote one seen HOI class as $s$ and the related seen text prompt token as $\hat{h_{T_s}}$ To identify the related seen class, we use text embeddings of the HOI class descriptions generated by the LLM, as explained in Text Prompt Design of Section 3.1. The related seen class is formulated as follows:

$$s = \underset{s \in \mathbb{S}}{\arg \max} \ (f_{\text{txt}_u})^T \cdot f_{\text{txt}_s} \tag{4}$$

Directly refining prompts for unseen classes based solely on selected seen classes may be insufficient due to inherent differences between the seen and unseen classes. Thus, we propose utilizing disparity information, defined as the differences between unseen class $u$ and seen class $s$, as provided by the LLM. The architecture of UTPL is shown in Fig. 3. Please refer to Appendix Section 7.3 for detailed explanation with examples.

Since the description can be too long to be encoded at once, we process each sentence separately by using the CLIP text encoder. Thus, set the text embedding of disparity descriptions for an unseen HOI class $u$ as $f_{\text{txt}_u}^{\text{disp}} \in \mathcal{R}^{m*d_t}$, where $m$ is the number of sentences in the text description. Then we can compute the refined learnable text prompts $\tilde{h_{T_u}}$ for unseen class $u$ as:

$$\tilde{h_{T_u}} = W_{up} \cdot \text{MHCA}(Q : W_{down} \cdot \hat{h_{T_u}}; \ K, V : W_{down} \cdot \text{concat}(f_{\text{txt}_u}^{\text{disp}}, \hat{h_{T_s}}, \hat{h_{T_u}})) + \hat{h_{T_u}}, \tag{5}$$

where $\mathrm{concat}$ indicates concatenation. For seen classes, $\tilde{h_{T_s}} = \hat{h_{T_s}}$. Although ground-truth labels for the unseen classes are not available during training, we propose two strategies to update $\tilde{h_{T_u}}$ in Eq. 5. First, we design a class-relation loss (refer to Eq. 16 in the Appendix) to keep the relationship between seen and unseen classes, measured by cosine similarity between text features. As a result, unseen prompts can be refined based on their relation to seen classes. Second, the annotated training data serves as negative samples for unseen HOIs. If the prediction score for an unseen class is too high, the model is penalized (Eq. 17 in the Appendix).

### 3.3 Deep Visual-Text Prompt Learning

We denote a set of learnable text prompts as $\mathcal{H}_\mathcal{T} = [\tilde{h_T^1}, \tilde{h_T^2}, \cdots, \tilde{h_T^N}]$ and learnable visual prompts as $\mathcal{H}_\mathcal{V} = [\hat{h_V^1}, \hat{h_V^2}, \cdots, \hat{h_V^N}]$. $\tilde{h_T^i} \in \mathcal{R}^{C*p*d_t}, \hat{h_V^i} \in \mathcal{R}^{p*d_v}, i = 1, 2, \cdots, N$. $N$ means we intend to introduce new learnable prompts from the first layer to layer $N$.

**Deep Text Prompt Learning** The text encoder is composed of $K$ transformer layers $\{\mathcal{T}_i\}_{i=1}^K$. The CLIP text encoder generates text features by tokenizing the words and converting them into word embeddings $W_0 = [w_0^1, w_0^2, \cdots, w_0^P] \in \mathcal{R}^{P*d_t}$. Text features $W_i$ will be processed by $i^{\mathrm{th}}$ layer of the text transformer:

$$
\begin{aligned}
[W_{i+1}, \_] &= \mathcal{T}_i(W_i, \tilde{h_T^i}), \ i = 1, 2, \cdots, N, \\
[W_{i+1}, \tilde{h_T^{i+1}}] &= \mathcal{T}_i(W_i, \tilde{h_T^i}), \ i = N+1, N+2, \cdots, K.
\end{aligned}
\tag{6}
$$

The final text representation $W_t$ can be obtained by:

$$
W_t = \mathrm{TextProj}(W_K),
\tag{7}
$$

where $W_t \in \mathcal{R}^{1*d_a}$ and $d_a$ is the feature dimension of the final aligned text and visual features in CLIP. Since our learnable text prompt $\tilde{h_T^i}$ is specific for each HOI class, the whole text representation can be obtained $\mathcal{W}_t = [W_{t_1}, W_{t_2}, \cdots, W_{t_C}]$.

**Deep Visual Prompt Learning** The visual encoder is also composed of $K$ transformer layers $\{\mathcal{V}_i\}_{i=1}^K$. After each layer $\mathcal{V}_i$, there is an adapter $\mathcal{A}_i$ to inject object position and category information [27]. CLIP visual encoder splits image $\mathcal{I}$ into $D = d_p * d_p$ fixed-size patches, which are projected to obtain visual features $E_1 \in \mathcal{R}^{D*d_v}$. Visual features $E_i$ are processed by $i^{\mathrm{th}}$ layer of the visual transformer:

$$
\begin{aligned}
[c_{i+1}, E_{i+1}, \_] &= \mathcal{A}_i(\mathcal{V}_i(c_i, E_i, \hat{h_V^i})), \ i = 1, 2, \cdots, N, \\
[c_{i+1}, E_{i+1}, \hat{h_V^{i+1}}] &= \mathcal{A}_i(\mathcal{V}_i(c_i, E_i, \hat{h_V^i})), \ i = N+1, N+2, \cdots, K,
\end{aligned}
\tag{8}
$$

where $c_i$ is the class token for the $i^{\mathrm{th}}$ layer.

The final visual representation $E_v \in \mathcal{R}^{1*d_a}$ can be computed as follows:

$$
E_v = \mathrm{VisualProj}(E_K).
\tag{9}
$$

$E_v \in \mathcal{R}^{D*d_v}$ can be re-sized to $E_v \in \mathcal{R}^{d_p*d_p*d_v}$. Since off-the-shelf detectors can provide object detection results, we select bounding boxes with confidence score $sc > \theta$ and apply RoI-Align [13] to obtain features for each detected human and object $E_{\mathrm{hum}}, E_{\mathrm{obj}}$. Then, the intra-HOI feature fusion extracts HOI feature $E_{\mathrm{hoi}} \in \mathcal{R}^{1*d_a}$ from $E_{\mathrm{hum}}, E_{\mathrm{obj}}$:

$$
E_{\mathrm{hoi}} = MLP(E_{\mathrm{hum}}, E_{\mathrm{obj}}),
\tag{10}
$$

where $MLP$ represents multi-layer perception. In any given image $\mathcal{I}$, multiple humans and objects may be present, leading to the detection of several potential human-object pairs. Assume there are $q$ human-object (H-O) pairs in image $\mathcal{I}$ and denote all HOI features as $\mathcal{E}_{\mathrm{hoi}}^{\mathcal{I}} = [E_{\mathrm{hoi}}^1, E_{\mathrm{hoi}}^2, \cdots, E_{\mathrm{hoi}}^q]$. Incorporating surrounding HOI features enriches context information for each HOI visual feature by capturing additional human-object relational details and interactions, thereby enhancing HOI features. We name this process inter-HOI feature fusion. Thus, final visual representation $\tilde{\mathcal{E}}_{\mathrm{hoi}}^{\mathcal{I}}$ is obtained by:

$$
\tilde{\mathcal{E}}_{\mathrm{hoi}}^{\mathcal{I}} = W_{\mathrm{up}} \cdot \mathrm{MHSA}(W_{\mathrm{down}} \cdot \mathcal{E}_{\mathrm{hoi}}^{\mathcal{I}}) + \mathcal{E}_{\mathrm{hoi}}^{\mathcal{I}},
\tag{11}
$$

where MHSA means multi-head self-attention [50] and $\tilde{\mathcal{E}}_{\mathrm{hoi}}^{\mathcal{I}} = [\tilde{E}_{\mathrm{hoi}}^1, \tilde{E}_{\mathrm{hoi}}^2, \cdots, \tilde{E}_{\mathrm{hoi}}^q]$. Then, we can calculate the prediction for each H-O pair $\mathrm{hoi}_i$.

$$
p_{\mathrm{hoi}}(c|\mathrm{hoi}_i) = \frac{\exp(\tilde{E}_{\mathrm{hoi}}^i \cdot (W_{t_c})^T)}{\sum_{k=1}^C \exp(\tilde{E}_{\mathrm{hoi}}^i \cdot (W_{t_k})^T)}, \ c = 1, 2, \cdots, C.
\tag{12}
$$

Please refer to Appendix Section 7.5 for more details for training and inference.

# 4 Experiments

Table 1: Unseen-verb (UV) comparison on HICO-DET with state-of-the-art methods. * indicates the model size is estimated according to papers [41, 4]. "TP" denotes the trainable parameters.

| Method | Setting | Backbone | TP | mAP | | |
| | | | | Full | Unseen | Seen |
|---|---|---|---|---|---|---|
| GEN-VLKT (CVPR'22) [33] | UV | Resnet50+ViT-B | 42.05M | 28.74 | 20.96 | 30.23 |
| EoID (AAAI'23) [51] | UV | Resnet50 | 41.45M | 29.61 | 22.71 | 30.73 |
| HOICLIP (CVPR'23) [44] | UV | Resnet50+ViT-B | 66.18M | 31.09 | 24.30 | 32.19 |
| CLIP4HOI (NeurIPS'23) [41] | UV | Resnet50+ViT-B | 56.7M* | 30.42 | **26.02** | 31.14 |
| Ours | UV | Resnet50+ViT-B | **6.85M** | **32.32** | 25.10 | **33.49** |
| UniHOI (NeurIPS'23) [4] | UV | Resnet50+ViT-L | 52.3M* | 34.68 | 26.05 | 36.78 |
| Ours | UV | Resnet50+ViT-L | **14.07M** | **36.84** | **28.82** | **38.15** |

Table 2: Rare-first unseen-composition (RF-UC) and Nonrare-first unseen composition (NF-UC) comparison on HICO-DET with state-of-the-art methods.

| Method | Backbone | Setting | mAP | | | Setting | mAP | | |
| | | | Full | Unseen | Seen | | Full | Unseen | Seen |
|---|---|---|---|---|---|---|---|---|---|
| GEN-VLKT [33] | Resnet50+ViT-B | RF | 30.56 | 21.36 | 32.91 | NF | 23.71 | 25.05 | 23.38 |
| EoID [51] | Resnet50 | RF | 29.52 | 22.04 | 31.39 | NF | 26.69 | 26.77 | 26.66 |
| HOICLIP [44] | Resnet50+ViT-B | RF | 32.99 | 25.53 | 28.47 | NF | 27.75 | 26.39 | 28.10 |
| ADA-CM [27] | Resnet50+ViT-B | RF | 33.01 | 27.63 | 34.35 | NF | **31.39** | 32.41 | **31.13** |
| CLIP4HOI [41] | Resnet50+ViT-B | RF | **34.08** | 28.47 | **35.48** | NF | 28.90 | 31.44 | 28.26 |
| Ours | Resnet50+ViT-B | RF | 33.13 | **29.02** | 34.15 | NF | 31.17 | **33.66** | 30.55 |
| UniHOI [4] | Resnet50+ViT-L | RF | 32.27 | 28.68 | 33.16 | NF | 31.79 | 28.45 | 32.63 |
| Ours | Resnet50+ViT-L | RF | **36.73** | **34.24** | **37.35** | NF | **34.84** | **36.33** | **34.47** |

## 4.1 Zero-Shot HOI Setting Definition

Our method follows the existing zero-shot HOI setting, which involves predicting unseen HOI classes and typically includes using unseen class names during training [14, 15, 16, 44, 51]. In particular, VCL, FCL and ATL [14, 16, 15] "compose novel HOI samples" during training with the unseen (novel) HOI class names. EoID [51] distills CLIP "with predefined HOI prompts" including both seen and unseen class names. HOICLIP [44] introduces "verb class representation" during training, including both seen and unseen classes.

## 4.2 Implementation Details

We evaluate our method on HICO-DET by following the established protocol of zero-shot two-stage HOI detection methods [14, 3, 27]. Our object detector utilizes a pre-trained DETR model [5] with a ResNet50 backbone [12]. As for our learnable prompts design, we set $p = 2, N = 9$. The LLaVA-v1.5-7b model [37] is used to provide text description, as explained in Section 3.1 and 3.2. For all experiments, our batch size is set as 16 on 4 Nvidia A5000 GPUs. We use AdamW [39] as the optimizer and the initial learning rate is 1e-3. For more implementation details, please refer to Appendix Section 7.1.

## 4.3 Comparison with State-of-the-Art Zero-Shot HOI Methods

**Unseen-Verb Setting** In Table 1, we compare our method to existing zero-shot HOI detection approaches under the unseen-verb zero-shot setting. The results demonstrate that our method not only achieves competitive performance but also requires only **10.35%** to **33.95%** of the trainable parameters compared to existing zero-shot HOI detection methods thanks to our novel prompt learning design. While our method shows a minor drop in performance compared to CLIP4HOI under the unseen verb setting, it is important to consider the significant reduction in trainable parameters that our method achieves—87.9% fewer than CLIP4HOI. In contrast, the existing HOI methods [33, 44, 41, 4] that align with VLMs unanimously require significantly more trainable

Table 3: Unseen-object (UO) comparison on HICO-DET with state-of-the-art methods. * indicates the model size is estimated according to papers [41, 4]. † denotes methods that use a DETR object detection model pretrained on HICO-DET. Results *without* † indicate the use of a DETR object detection model pretrained on MS-COCO. "TP" denotes the trainable parameters.

| Method | Setting | Backbone | TP | mAP | | |
| | | | | Full | Unseen | Seen |
| --- | --- | --- | --- | --- | --- | --- |
| FCL† (CVPR'21) [16] | UO | Resnet50 | - | 19.87 | 15.54 | 20.74 |
| ATL† (CVPR'21) [15] | UO | Resnet50 | - | 20.47 | 15.11 | 21.54 |
| GEN-VLKT (CVPR'22) [33] | UO | Resnet50 | 42.05M | 25.63 | 10.51 | 28.92 |
| HOICLIP (CVPR'23) [44] | UO | Resnet50+ViT-B | 66.18M | 28.53 | 16.20 | 30.99 |
| CLIP4HOI† (NeurIPS'23) [41] | UO | Resnet50+ViT-B | 56.7M* | **32.58** | 31.79 | **32.73** |
| Ours | UO | Resnet50+ViT-B | **6.85M** | 27.90 | 31.63 | 27.16 |
| Ours† | UO | Resnet50+ViT-B | **6.85M** | 32.27 | **33.28** | 32.06 |
| UniHOI (NeurIPS'23) [4] | UO | Resnet50+ViT-L | 52.3M* | 31.56 | 19.72 | 34.76 |
| Ours | UO | Resnet50+ViT-L | **14.07M** | 31.42 | 33.08 | 31.09 |
| Ours† | UO | Resnet50+ViT-L | **14.07M** | **36.38** | **38.17** | **36.02** |

model parameters (e.g., 42.05M $\sim$ 66.18M with ViT-B visual encoder), whereas our method operates efficiently with only 6.85M trainable parameters.

UniHOI [4] is the state-of-the-art method in the unseen verb (UV) setting. Compared to UniHOI which aligns HOI features to BLIP [29] text embeddings, our method achieves improved performance, especially in the unseen category with a 2.77 increase in mAP. At the same time, our model requires only 26.9% of the trainable parameters compared to UniHOI. This demonstrates the effectiveness and efficiency of our proposed prompt learning framework in adapting the VLM for zero-shot HOI tasks.

**Unseen-Composition Setting** In Table 2, we provide the zero-shot performance comparison in rare-first unseen-composition (RF-UC) and nonrare-first unseen-composition (NF-UC) settings. With the ViT-B visual encoder, our method establishes a new standard for unseen-class performance across both RF and NF settings, outperforming the previous state-of-the-art method, CLIP4HOI [41], while requiring only 12.08% of its trainable parameters. Compared to UniHOI with ViT-L visual encoder [4], our method surpasses it by 5.56 mAP in unseen performance of the RF-UC setting and by 7.88 mAP in the NF-UC setting with only 26.9% of the trainable parameters of UniHOI.

**Unseen-Object Setting** We provide the unseen-object setting comparison in Table 3. Following previous methods [44, 33], we employ a DETR object detector pre-trained on MS-COCO [34] for object detection. To keep consistent with CLIP4HOI [41] and FCL [16], we also provide the results by using the DETR model pre-trained on HICO-DET [27] (marked with † in Table 3). CLIP4HOI [41] is the state-of-the-art method in unseen performance. With the same object detector pre-trained on HICO-DET, our method shows improved performance for unseen class prediction, outperforming CLIP4HOI by 1.49 mAP, with only 12.08% of its trainable model parameters. UniHOI [4] is the state-of-the-art method in seen performance. With the same object detector as UniHOI, our method outperforms UniHOI in unseen category by 13.36 mAP.

## 4.4 Ablation Studies

We conduct the ablation study for our text and visual prompt learning design in Table 4. The first row shows the performance of our baseline, MaPLe [22]. Our intra-HOI fusion module improves the seen HOI performance by 7.41 mAP, as shown in the second row. In the third row, the inclusion of the visual adapter [27] enhances performance for seen classes while negatively affecting unseen HOI performance.

Comparing the third row and fourth row, LLM guidance, shown in Fig. 2, notably enhances unseen-class learning, resulting in a 1.52 mAP improvement. LLM guidance, by integrating learnable text prompts with detailed HOI class descriptions, improves the model's understanding of unseen classes, providing richer information compared to simple class names. Additionally, the inclusion of the UTPL module significantly boosts unseen class performance, with a **2.42** mAP improvement. Later, the inter-HOI fusion benefits both the seen and unseen learning by providing more context

Table 4: Ablation study of our method in zero-shot unseen verb setting on HICO-DET.

| Intra-HOI fusion | visual adapter [27] | LLM Guide | UTPL | Inter-HOI fusion | VLM Guide | mAP | | |
|:---:|:---:|:---:|:---:|:---:|:---:|:---:|:---:|:---:|
| | | | | | | Full | Unseen | Seen |
| ✗ | ✗ | ✗ | ✗ | ✗ | ✗ | 26.26 | 17.19 | 27.73 |
| ✓ | ✗ | ✗ | ✗ | ✗ | ✗ | 33.52 | 23.54 | 35.14 |
| ✓ | ✓ | ✗ | ✗ | ✗ | ✗ | 35.40 | 22.91 | 37.44 |
| ✓ | ✓ | ✓ | ✗ | ✗ | ✗ | 35.62 | 24.43 | 37.44 |
| ✓ | ✓ | ✓ | ✓ | ✗ | ✗ | 36.23 | 26.85 | 37.76 |
| ✓ | ✓ | ✓ | ✓ | ✓ | ✗ | 36.70 | 27.49 | 38.10 |
| ✓ | ✓ | ✓ | ✓ | ✓ | ✓ | **36.84** | **28.82** | **38.15** |

Table 5: Prompt learning evaluation in zero-shot unseen verb setting on HICO-DET.

| Method | mAP | | |
|:---:|:---:|:---:|:---:|
| | Full | Unseen | Seen |
| CLIP [47] | 13.18 | 13.96 | 13.05 |
| MaPLe [22] | 26.26 | 17.19 | 27.73 |
| MaPLe [22] + visual adapter [27] | 32.70 | 22.89 | 34.29 |
| Ours | **36.84** | **28.82** | **38.15** |

information for visual HOI features. Finally, the VLM guidance effectiveness is evaluated, which enhances the unseen performance by 1.33 mAP.

We also provide the experimental comparison with CLIP [47] and MaPLe [22] in Table 5. Since they are designed for general image classification, directly applying it to HOI tasks for comparison is unfair as they may not focus on the human-object region features. Therefore, we crop the union region from the original visual features for each human-object pair to obtain the HOI feature. HOI prediction is obtained by using cosine similarity between HOI and text features for each class.

As shown in Table 5, although with learnable prompts, MaPLe [22] outperforms CLIP on seen classes, its performance on unseen classes is far behind that of the seen classes, showing reduced generalization capability. Compared to our method, MaPLe lags significantly behind on unseen performance, with an 11.63 decrease in mAP. Additionally, the third row introduces another baseline that incorporates MaPLe with visual adapter (i.e., $\mathcal{A}_i$) [27], which is also used in our framework, ensuring a fair comparison. The performance improvement from the third row to the fourth row, with gains of 5.93 mAP for unseen classes and 3.86 mAP for seen classes, clearly demonstrates the effectiveness of our method in zero-shot HOI detection.

### 4.5 Qualitative Results

Fig. 4 shows the qualitative results of MaPLe [22] and our method in the unseen-verb zero-shot setting. In particular, MaPLe struggles to detect unseen classes, either missing unseen HOI classes or predicting the wrong unseen HOI classes. For example, if an image only contains unseen classes, MaPLe tends to predict wrong seen classes and miss the correct ones. As shown in the bottom right of Fig. 4, this image contains unseen class only ("wear tie"), MaPLe predicts related wrong seen classes such as "pulling tie" and "adjusting tie", but fails to predict the ground-truth unseen HOI ("wear tie"). This shows the limited generalization ability of MaPLe to unseen classes. In contrast to MaPLe, our method can predict both seen and unseen classes more accurately.

### 4.6 Fully Supervised Setting for HOI Detection

We conducted the fully-supervised experiments on both the HICO-DET and V-COCO benchmarks [6, 34]. Our method achieves a competitive 38.61 mAP on the HICO-DET dataset, with a smaller performance drop between rare and non-rare classes (1.19 mAP) compared to AGER [49] (4.18 mAP), the current state-of-the-art one-stage method. Our method also outperforms the best-performing zero-shot HOI method, CLIP4HOI [41], by 3.28 mAP on HICO-DET. On the V-COCO benchmark, our method achieves state-of-the-art performance with 66.2 AP in Scenario 2. Detailed discussions and comparisons are provided in the Appendix 7.6.

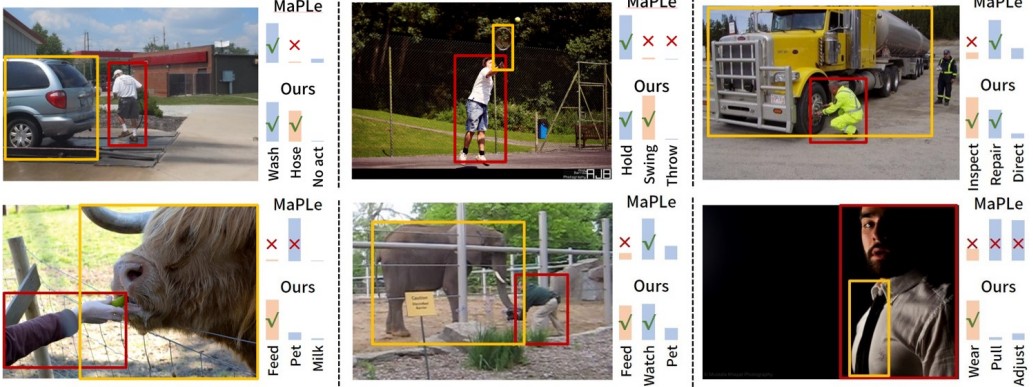

Figure 4: Qualitative comparison with MaPLe [22] for unseen-verb zero-shot HOI detection.The orange bar represents the unseen class prediction and the blue bar means the seen class prediction.

## 4.7 Discussion of Descriptors from LLMs

Descriptors, developed to leverage the LLMs for downstream tasks [42, 20], involve attribute decomposition to benefit category recognition. We adopt the attribute decomposition concept from DVDet [20] and tailor it to our EZ-HOI framework. Following DVDet, we generate action descriptors for each class and integrate them into HOI class text features, thus enhancing the detail and distinctiveness of class representations. Descriptors with low cosine similarity to the class text features are discarded to avoid noise. With our straightforward adoption of action descriptors, the result shows 0.31 mAP improvement compared to our EZ-HOI, achieving 32.63 mAP under the unseen-verb setting. This indicates that LLM-generated descriptions, such as action descriptors, hold potential to enhance HOI detection and are worth further exploration.

## 4.8 Limitations

Our method has some limitations. First, while our method significantly reduces the number of trainable parameters compared to existing approaches, it still requires fine-tuning on HOI-specific datasets such as HICO-DET. A training-free design, which could be used directly for HOI detection without the need for fine-tuning, would be more desirable. Second, zero-shot HOI detection requires the pre-definition of all HOI classes, both seen and unseen, following the current zero-shot HOI detection setting. This requirement restricts the generalizability to truly unseen classes. Consequently, our future work will aim to develop a robust open-category HOI detector that operates effectively without the need for predefined classes.

## 5 Conclusion

In this paper, we introduced the Efficient Zero-shot HOI Detection (EZ-HOI) approach, utilizing prompt learning to adapt visual-language models for zero-shot HOI detection. This method not only maintains competitive performance for unseen classes but also innovatively integrates learnable visual and text prompts. These prompts leverage foundation model information, thereby enriching prompt knowledge and enhancing the adaptability of VLMs. A significant challenge we addressed is the compromised performance on unseen classes, resulting from the training dataset containing only seen-class labels. To counter this, we developed a text prompt learning strategy that utilizes information from related seen classes to support the detection of unseen classes. We also employed an LLM to provide the nuanced differences between related seen and unseen classes, improving our method for unseen class prompt learning. Our method has demonstrated state-of-the-art performance across various zero-shot HOI detection settings while requiring only a third of the trainable parameters compared to existing methods.

# 6   Acknowledgements

This research/project is supported by the National Research Foundation, Singapore under its Strategic Capability Research Centres Funding Initiative. Any opinions, findings and conclusions or recommendations expressed in this material are those of the author(s) and do not reflect the views of National Research Foundation, Singapore.

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

# 7 Appendix

## 7.1 Implementation Details

Here we provide detailed implementation information. For the pre-trained CLIP model with the ViT-B visual encoder, the visual feature dimension $d_v = 768$, while the text feature dimension $d_t = 512$ and the final feature dimension for aligned visual and text features $d_a = 512$. For the CLIP model with the ViT-L visual encoder, $d_v = 1024, d_t = 768, d_a = 768$. We use an off-the-shelf object detector and add a threshold $\theta$ to filter out some low-confident predictions and we set $\theta = 0.2$. Since UniHOI [4] and CLIP4HOI [41] have not released their code, we estimate the trainable model parameters by modifying the HOICLIP [44] code according to the description in UniHOI and CLIP4HOI, which means that if some details are not mentioned in the UniHOI and CLIP4HOI papers, we use the HOICLIP design by default. In addition, we initialize the weights of all $W_{up}$ to 0, which stabilizes training by gradually fine-tuning the attention outputs in Eq. 2, Eq. 3, Eq. 5 and Eq. 11.

**Dataset and Evaluation Metrics**

We conducted extensive experiments using the widely-recognized HICO-DET [6] dataset for zero-shot HOI detection. HICO-DET comprises a total of 47,776 images, divided into 38,118 training images and 9,658 test images. This dataset features 600 Human-Object Interaction (HOI) classes, which are combinations derived from 117 action categories and 80 object categories. Our model's performance was evaluated in four distinct zero-shot HOI detection settings, categorized by the criterion for selecting the unseen HOI classes: rare-first unseen composition (RFUC), nonrare-first unseen composition (NFUC), unseen object (UO), and unseen verb (UV). These settings align with methodologies from previous research [33, 44, 41, 4].

Additionally, we also evaluate our model in the fully supervised setting on both the HICO-DET dataset and the V-COCO [34] dataset. V-COCO is a subset of COCO, with 10,396 images—split into 5,400 train-val images and 4,946 test images, and it encompasses 24 action classes and 80 object classes. Following standard evaluation protocols, we assessed our model using mean average precision (mAP) on the HICO-DET benchmark, while Average Precision (AP) in both Scenario 1 and Scenario 2 is used on the V-COCO benchmark [63, 44]. A prediction was deemed a true positive if the HOI classification was accurate and the Intersection over Union (IoU) between the predicted human and object bounding boxes and the ground-truth bounding boxes exceeded 0.5 [33, 27].

## 7.2 HOI Class Description

We utilize the LLM to provide detailed HOI class descriptions to guide the text prompt learning and we provide an example for detailed illustration. We take the HOI class "Swing a baseball bat" as an example, which is one unseen HOI class in the unseen-verb zero-shot setting on the HICO-DET benchmark. The generated HOI class description is

*"Swinging a baseball bat" describes a person using a baseball bat to hit a ball. This action typically involves the person holding the bat with both hands, standing in a stance with their feet shoulder-width apart, and using their body rotation to contact the ball.*

## 7.3 Disparity Information for UTPL Module

The disparity information mentioned in Section 3.2 aims to explore the difference between the unseen class and the related seen class. Here we demonstrate it with a specific example. We take the unseen class " hose a dog" as an example and its selected seen class is "wash a dog".The input text prompt for LLM to acquire the disparity description between the two classes is designed as " *Describe the definition of the phrase: hosing a dog and please focus on the attributes different from another phrase: washing a dog.* " Then we can obtain the detailed disparity description from LLM as

*The phrase "a person hosing a dog" refers to the action of washing a dog using a hose. This action is different from "a person washing a dog" as the person is using a hose to clean the dog, rather than simply using their hands to wash the dog. The use of a hose adds an additional element of water pressure and flow, which can make the cleaning process more efficient and effective.*

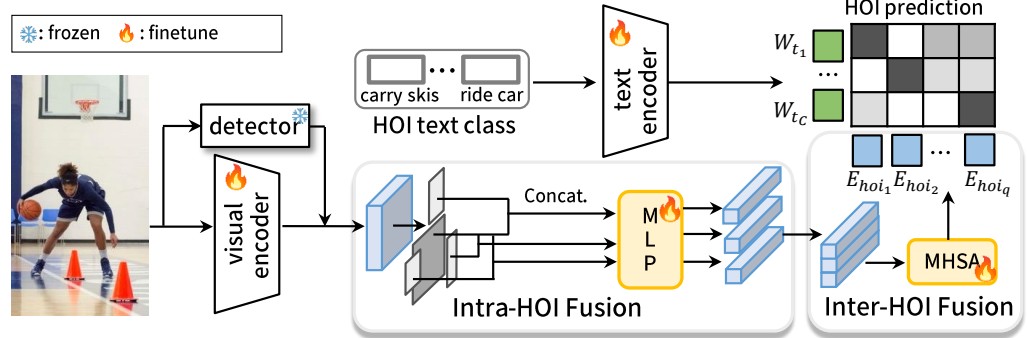

Figure 5: Detailed architecture for HOI feature fusion design. Intra-HOI feature fusion aims to extract HOI features from possible human region and object region features. Inter-HOI feature fusion aims to enhance the HOI features by incorporating the surrounding HOI feature context. "MHSA" refers to multi-head self-attention.

## 7.4   HOI Feature Fusion

As shown in Fig. 5, we also provide the detailed architecture of the intra- and inter-HOI feature fusion design. The intra-HOI feature fusion integrates human and object features for each human-object pair. To enrich context information, all HOI features are processed together using multi-head self-attention (MHSA), allowing each feature to become aware of its surroundings. This enhancement improves the context and overall performance of each HOI feature.

## 7.5   Training and Inference

Since $\mathcal{W}_t \in \mathcal{R}^{C*d_t}$ related to the number of total HOI classes, if the number is too large, the learnable prompts can be computationally expensive. Thus, we only select part of HOI classes and do action classification. Later, by combining action prediction with object detection results, we can obtain HOI prediction finally. Specifically, we select two HOI classes for each action with the following equation:

$$\text{hoi}_i, \text{hoi}_j = \underset{0<i,j\leq C}{\arg\min} f_{txt_i} \cdot (f_{txt_j})^T, \tag{13}$$

which means we select two HOI classes containing the same action with the most different semantic meanings. Since some actions can have multiple interpretations in different contexts (e.g., "hold apple" vs. "hold sheep"), randomly choosing two HOI classes with the same action does not cover the comprehensive meanings of the action. By selecting the most different HOI classes, we aim to capture a richer range of information for the action. Then, We calculate the HOI prediction score for $i^{\text{th}}$ H-O pair by:

$$p_{\text{hoi}}(c|\text{hoi}_i) = \frac{\exp(\tilde{E}^i_{\text{hoi}} \cdot (W_{t_c})^T)}{\sum_{k=1}^{C} \exp(\tilde{E}^i_{\text{hoi}} \cdot (W_{t_k})^T)}, \ c = 1, 2, \cdots, C. \tag{14}$$

The action scores can be obtained by:

$$s_a = p_{\text{hoi}} * \tilde{l}_{\text{align}}, \tag{15}$$

where $\tilde{l}_{\text{align}} \in \mathcal{R}^{C*d_a}$ means the action label for each selected HOI class. We adopt focal loss [35] $l_{focal}$ to train the action prediction. We also design a class-relation loss shown in the following equation:

$$l_{\text{relation}} = D_{KL}[\text{sim}(\mathcal{W}_t, \mathcal{W}_t) || \text{sim}(\mathcal{F}_{\text{txt}}, \mathcal{F}_{\text{txt}})],$$
$$\text{sim}(X, Y) = X^T \cdot Y \tag{16}$$

to keep the relation between adapted text class embeddings $W_t$ close to the original text embeddings of the HOI class description $f_{\text{txt}}$. $D_{KL}[\cdot || \cdot]$ denotes the KL divergence. Therefore, the training loss $L_{\text{train}}$ is computed by:

$$L_{\text{train}} = l_{\text{focal}} + \alpha l_{\text{relation}}, \tag{17}$$

and $\alpha$ is a hyperparameter. In the inference stage, we can obtain the HOI score for each human-object pair by:

$$s_{h,o}^a = (s_h * s_o)^\tau * \sigma(s_a), \tag{18}$$

where $s_h, s_o$ are the object detection confidence scores for humans and objects. $\tau$ is a hyperparameter. In Eq. 17, $\alpha = 150$, and in Eq. 18, $\tau = 1$ during training and $\tau = 2.8$ during inference [58, 59].

### 7.6 Quantitative Results

**Fully Supervised HOI Detection Setting** We conduct experiments under the fully supervised HOI detection setting on both the HICO-DET [6] and V-COCO [34] dataset, as shown in Table 6. In the UTPL module, we utilize common class prompts as inputs. This approach enables common HOIs to learn from rare ones, fostering a better differentiation between them. This strategy helps to alleviate overfitting to common HOI classes, thereby improving performance for both seen and unseen classes. As shown in Table 6, the performance drop between rare and non-rare classes ( 1.19 mAP) is much smaller than AGER (4.18 mAP), the best performance among the one-stage method. Additionally, our method achieves the best performance among the two-stage methods on the HICO-DET [6] dataset, outperforming CLIP4HOI [41] by 3.28 mAP.

As for the V-COCO benchmark, we achieve competitive performance among the two-stage methods with 66.2 AP performance under Scenario 2 evaluation. However, the smaller number of verb classes in V-COCO (24 classes), which have weaker connections (i.e., jumping vs. skateboarding skateboard), compared to HICO-DET (117 verb categories), which exhibits stronger connections (i.e., jumping vs. flipping skateboard), limits the potential of our method. Our UTPL design requires the learnable prompts to extract information from prompts of other classes, which also aids in better differentiation between similar classes. Due to the weakly connected classes in V-COCO, our UTPL design cannot work effectively. Despite this, we achieve performance comparable to CLIP4HOI [41], further demonstrating the effectiveness of our method.

Table 6: State-of-the-art Comparison on HICO-DET and V-COCO in the fully-supervised setting. **Bold** highlights the best-performing method within each of the two groups: one-stage and two-stage methods.

| Method | HICO-DET | | | V-COCO | |
|---|---|---|---|---|---|
| | Full | Rare | Nonrare | $AP_{role}^{S_1}$ | $AP_{role}^{S_2}$ |
| One-stage Methods | | | | | |
| GEN-VLKT (CVPR'22) [33] | 33.75 | 29.25 | 35.10 | 62.4 | 64.5 |
| HOICLIP (CVPR'23) [44] | 34.69 | 31.12 | 35.74 | 63.5 | 64.8 |
| RLIPv2 (ICCV'23) [56] | 35.38 | 29.61 | 37.10 | **65.9** | 68.0 |
| AGER (ICCV'23) [49] | **36.75** | **33.53** | **37.71** | 65.7 | **69.7** |
| LogicHOI (NeurIPS'23) [31] | 35.47 | 32.03 | 36.22 | 64.4 | 65.6 |
| Two-stage Methods | | | | | |
| UPT (CVPR'22) [59] | 32.62 | 28.62 | 33.81 | 59.0 | 64.5 |
| ADA-CM (ICCV'23) [27] | 38.40 | 37.52 | 38.66 | 58.6 | 64.0 |
| CLIP4HOI (NeurIPS'23) [41] | 35.33 | 33.95 | 35.75 | - | **66.3** |
| Ours | **38.61** | **37.70** | **38.89** | **60.5** | 66.2 |

### 7.7 Ablation Studies

Table 7 shows the ablation study for the hyperparameter $N$, where $N$ means we introduce learnable prompts from the first layer until layer $N$ in both visual and text encoders. We use N=9 in our main paper because it shows the best unseen performance.

Table 8 shows the ablation study for the different positions of learnable prompts. Position $i - j$ means we insert learnable prompts from layer $i$ to layer $j$ in both the text encoder and visual encoder. Here we always insert learnable prompts into 9 layers and only change the position. We find that the position of learnable prompts does not affect the outcome too much. Thus, we follow [22] and fine-tune layers 1-9, which show slightly better unseen performance.

Table 7: Ablation study for hyperparameter N under the unseen-verb setting.

| N | mAP | | |
|---|---|---|---|
| | Full | Unseen | Seen |
| 4 | 32.60 | 24.10 | 33.99 |
| 9 | 32.32 | **25.10** | 33.49 |
| 12 | 32.76 | 23.98 | 34.19 |

Table 8: Ablation study for hyperparameter N under the unseen-verb setting.

| Position | mAP | | |
|---|---|---|---|
| | Full | Unseen | Seen |
| 1-9 | 32.32 | 25.10 | 33.49 |
| 3-11 | 32.24 | 24.83 | 33.45 |
| 4-12 | 32.40 | 24.82 | 33.64 |

### 7.8 Qualitative Results

As shown in Fig. 6, we provide more qualitative results in three zero-shot HOI settings: unseen-verb, rare-first unseen-composition, and nonrare-first unseen-composition settings. Compared to MaPLe [22], our method obtains better generalization capability to unseen HOI classes owing to our efficient prompt learning design.

In Fig. 7, we show more qualitative comparisons to illustrate the effectiveness of the LLM guidance and UTPL design. The baseline here refers to the method in the main paper's third row of Table 4. LLM guidance design utilizes the general description from LLM for each HOI class and the UTPL module integrates the distinctive description from LLM. As shown in Fig. 7, the LLM guidance provides detailed class information, improving the performance over the baseline. Distinctive descriptions from UTPL design help to distinguish unseen classes from related seen HOIs, enhancing unseen performance and challenging case predictions.

### 7.9 Discussion for Zero-Shot HOI Detection Definition

Our method follows the standard zero-shot HOI setting, where unseen class names are used during training [14, 15, 16, 51, 44], as discussed in Section 4.1. Specifically, before training, the whole verb set $\mathbb{V} = \{\mathbb{V}_{seen}, \mathbb{V}_{unseen}\}$ and the whole object set $\mathbb{O} = \{\mathbb{O}_{seen}, \mathbb{O}_{unseen}\}$ are all pre-defined. The four settings of zero-shot HOI detection include : 1) Rare-First Unseen Composition (RF-UC), where for all $\text{hoi}_i = (v_i, o_i) \in \mathbb{U}$, we have $v_i \in \mathbb{V}_{seen}, o_i \in \mathbb{O}_{seen}$ and $\text{hoi}_i$ appears less than 10 times in the training set, belonging to rare HOI classes. 2) Nonrare-First Unseen Composition, where for all $\text{hoi}_i = (v_i, o_i) \in \mathbb{U}$, we have $v_i \in \mathbb{V}_{seen}, o_i \in \mathbb{O}_{seen}$ and $\text{hoi}_i$ appears more than 10 times in the training set, belonging to nonrare HOI classes. 3) Unseen Verb (UV), where $\text{hoi}_i = (v_i, o_i) \in \mathbb{U}$, we have $v_i \in \mathbb{V}_{unseen}, o_i \in \mathbb{O}_{seen}$. 4) unseen Object (UO), where $\text{hoi}_i = (v_i, o_i) \in \mathbb{U}$, we have $v_i \in \mathbb{V}_{seen}, o_i \in \mathbb{O}_{unseen}$.

Beyond the zero-shot HOI setting, there are HOI unknown concept discovery [17] and open-vocabulary HOI detection [28], where unseen class names cannot be used in training. The open-vocabulary setting differs from HOI unknown concept discovery, with a much wider range of unseen HOI classes during testing.

### 7.10 Discussion of Broader Impacts

Our approach to zero-shot HOI detection reduces reliance on extensive annotated datasets, enhancing accessibility for organizations with limited resources and promoting inclusivity in technology adoption. This technology could notably improve assistive devices, offering more intuitive aids for individuals with disabilities by enabling better understanding of new environments.

However, these advancements also pose risks. The capability to interpret human-object interactions could be used for surveillance, potentially infringing on privacy. Additionally, reliance on existing visual-language models may maintain embedded biases, leading to discriminatory outcomes. To

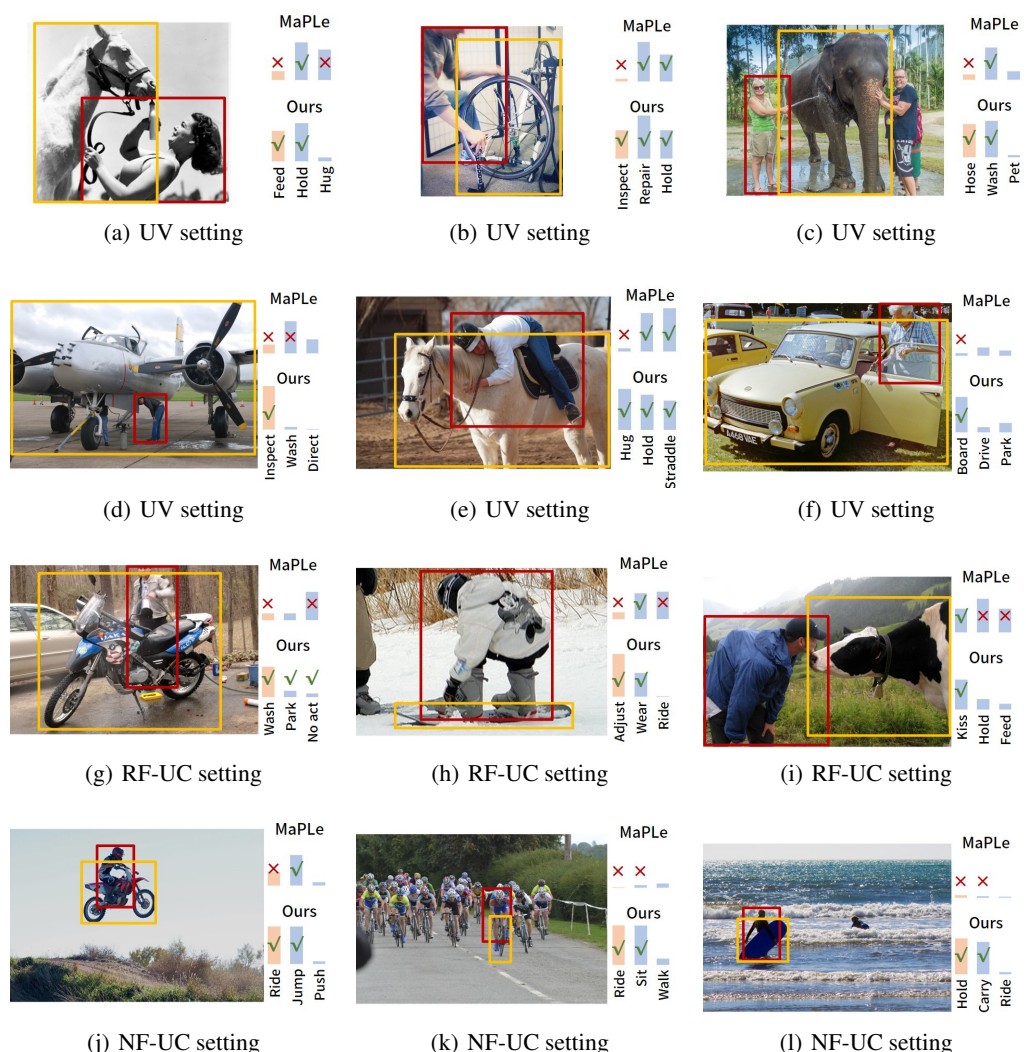

Figure 6: Qualitative comparison of zero-shot HOI detection between our method and MaPLe [22]. We use orange color to represent unseen HOI classes and use blue color for seen ones. For images containing multiple HOI results, we only present one prediction for clearer demonstration and comparison.

address these concerns, we recommend developing rigorous ethical guidelines and governance frameworks to regulate the deployment of HOI detection technologies, alongside efforts to identify and mitigate biases in foundational models.

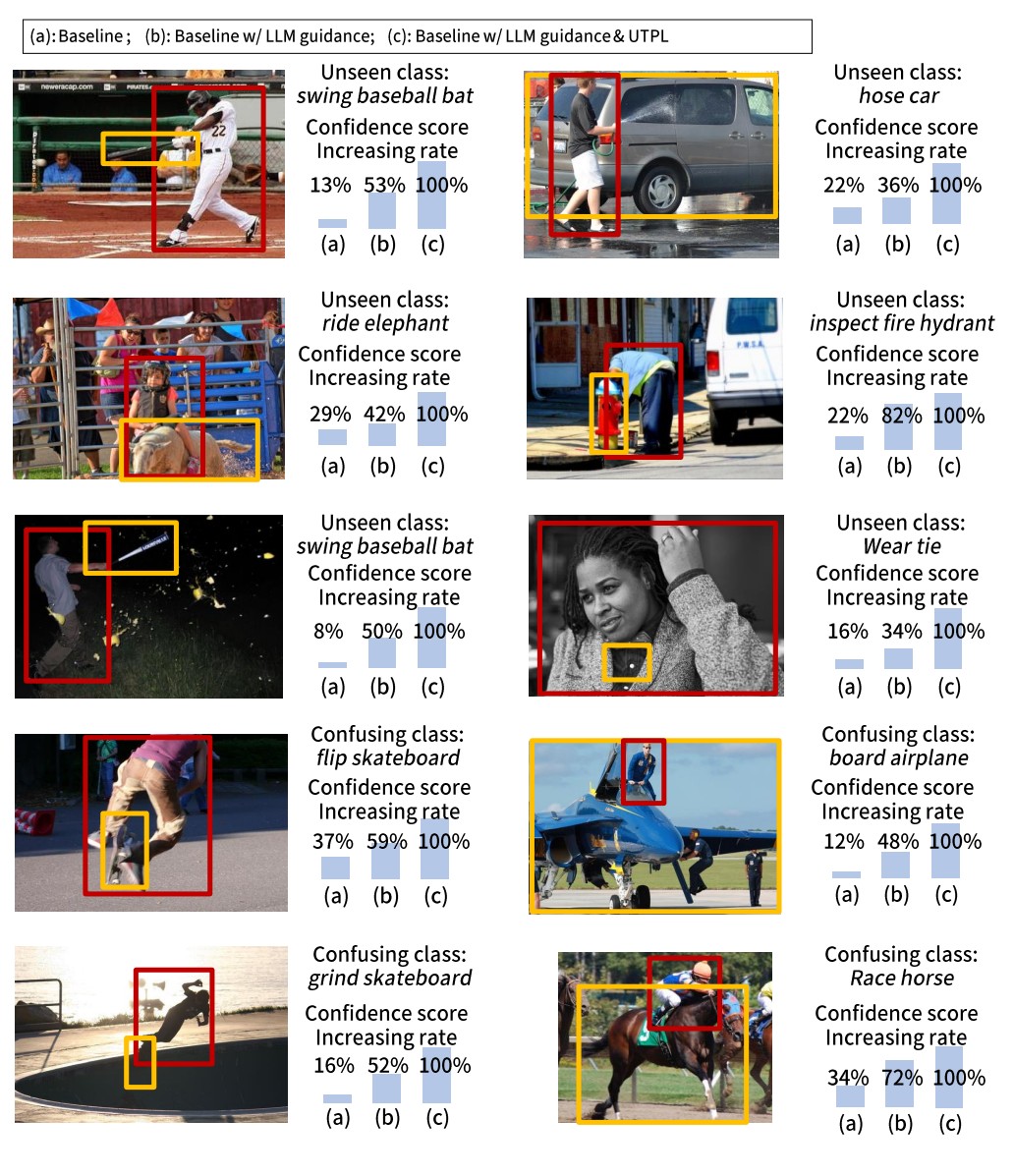

Figure 7: Qualitative results to show the effectiveness of the LLM guidance and UTPL design. We conduct the comparison under the unseen-verb zero-shot HOI setting.

