# OpenReview forum: "EZ-HOI: VLM Adaptation via Guided Prompt Learning for Zero-Shot HOI Detection"
_NeurIPS.cc/2024/Conference — NeurIPS 2024 poster_

### Official Review · Reviewer_f78r · 2024-06-25

**Soundness:** 3
**Presentation:** 3
**Contribution:** 4
**Rating:** 7
**Confidence:** 4

**Summary:**

In order to solve the HOI detection problem in a zero-shot setting, the authors propose the Unseen-class Text Prompt Learning module. Using learnable visual prompts and textual prompts, it effectively utilizes the knowledge from large language models and Vision-Language Models and performs well in unseen class HOI detection.

**Strengths:**

1.The work is innovative, well-written and has clear diagrams.
2.The authors' proposed learnable prompts scheme and UTPL module are novel.

**Weaknesses:**

1. In Qualitative Results, the authors use Figure 4 for visual illustration, but it is not possible to conclude from Figure 4 that "MaPLe tends to predict seen classes with high confidence scores". The authors should have given a clearer and more accurate illustration of the figure.
2. The authors claim to have "employed an LLM to provide the nuanced differences between related seen and unseen classes, improving our method for unseen class prompt learning. prompt learning." However, the information provided by the LLM may be misleading or inaccurate, did the authors filter the information? If not, what can be done to avoid HOI detection being negatively affected from the information provided by LLM?
3. For the UTPL module, the authors split the LLM-generated description into multiple sentences; how do they control the length of individual statements?
4. The author mentions: " A prediction was deemed a true positive if the HOI classification was accurate and the Intersection over Union (IoU) between the predicted human and object bounding boxes and the ground-truth bounding boxes exceeded 0.5.", why is the threshold value chosen to be 0.5 instead of 0.7 or other? What is the basis for the choice?
5. WRITING DETAILS: Abbreviated nouns should be introduced the first time they appear.

**Questions:**

1. In Qualitative Results, the authors use Figure 4 for visual illustration, but it is not possible to conclude from Figure 4 that "MaPLe tends to predict seen classes with high confidence scores". The authors should have given a clearer and more accurate illustration of the figure.
2. The authors claim to have "employed an LLM to provide the nuanced differences between related seen and unseen classes, improving our method for unseen class prompt learning. prompt learning." However, the information provided by the LLM may be misleading or inaccurate, did the authors filter the information? If not, what can be done to avoid HOI detection being negatively affected from the information provided by LLM?
3. For the UTPL module, the authors split the LLM-generated description into multiple sentences; how do they control the length of individual statements?
4. The author mentions: " A prediction was deemed a true positive if the HOI classification was accurate and the Intersection over Union (IoU) between the predicted human and object bounding boxes and the ground-truth bounding boxes exceeded 0.5.", why is the threshold value chosen to be 0.5 instead of 0.7 or other? What is the basis for the choice?
5. WRITING DETAILS: Abbreviated nouns should be introduced the first time they appear.

**Limitations:**

Yes

---

> ### Author Rebuttal · Authors · 2024-08-07
>
> We appreciate the positive feedback and detailed reviews. The following is our response for your questions. Note all references are from the main paper’s citations.
>
> ## 1. Illustration for Fig. 4
> We provide more illustration for Fig. 4 in the following.
> Fig. 4 shows the qualitative results of both our method and MaPLe. In particular, MaPLe struggles to detect unseen classes, either missing unseen HOI classes or predicting the wrong unseen HOI classes. For example, if an image only contains unseen classes, MaPLe tends to predict wrong seen classes and miss the correct ones. As shown in the bottom right of Fig. 4, this image contains unseen class only ("wear tie"), MaPLe predicts related wrong seen classes such as "pulling tie" and "adjusting tie", but fails to predict the ground-truth unseen HOI ("wear tie").
>
> This shows the limited generalization ability of MaPLe to unseen classes. In contrast to MaPLe, our method can predict both seen and unseen classes more accurately. We will revise the discussion for Fig. 4 in the updated paper.
>
> ## 2. Avoid negative effect of misleading information from LLM in UTPL
> We agree that LLM can provide misleading information.
> To deal with the problem, we utilize LLM description information with learnable attention.
> Specifically, the UTPL module integrates the LLM information into the unseen text learnable prompts, through multi-head cross-attention (MHCA), where learnable attention is optimized during the training.
> The training supervision for UTPL is from the class-relation loss in Eq. (15), to retain the relationship among each HOI class, indicated by cosine similarity between text class features.
> If a wrong information is provided by LLM, then the unseen text features, integrated with the wrong information, display improper relations to other seen class text features. This will be penalized by the class-relation loss, and then the attention to the wrong information will be reduced in UTPL for unseen prompt learning.
> Therefore, the design can mitigate the possible negative effect from inaccurate information provided by LLM
>
> In the future, we will also explore other approaches such as chain-of-thought (CoT) and retrieval-augmented generation (RAG) to mitigate the negative effect from LLM-generated misleading information.
>
> ## 3. Control the length of individual statements in UTPL
> The length control of individual statements is achieved by giving the prompt for LLM before  the generation for the desired description: _"Please reply with multiple short sentences in the following, instead of a long sentence."_
>
> ## 4. IoU threshold value selection
> We follow existing papers [19,28,38,23] to choose the threshold value, the IoU between predicted human and object bounding boxes and the ground-truth bounding boxes, to be 0.5.
> The threshold value is one criterion to judge whether a prediction is true positive or not. This is not only used in the evaluation during testing time, but also used in training to match predictions with ground-truth human-object pairs for model learning.
>
> We conduct the ablation study for the threshold value selection in training time, as shown in the following table. We find that using 0.5 achieves the best performance for both seen and unseen classes.
>
> | Threshold | Full | Unseen | Seen |
> | :-----| :----: | :----: |:----: |
> | 0.2 |31.02 |23.05| 32.32|
> |0.5|32.32|25.10|33.49|
> |0.7| 29.59| 23.66 | 30.55|
>
> ## Writing details
>
> We will make sure the abbreviations are introduced with their full names the first time they appear in the revised version of our paper.

---

> > ### Comment · Area_Chair_VWco · 2024-08-12
> > **Please read through rebuttal, and see if the rebuttal has addressed your concerns or you have further comments.**
> >
> > Hi dear reviewer,
> >
> > Please read through rebuttal, and see if the rebuttal has addressed your concerns or you have further comments.
> >
> > Thanks,
> >
> > AC

---

### Official Review · Reviewer_kLJ1 · 2024-07-10

**Soundness:** 3
**Presentation:** 3
**Contribution:** 2
**Rating:** 5
**Confidence:** 3

**Summary:**

This work presents EZ-HOI, an innovative framework that addresses the zero-shot HOI detection challenge by employing prompt learning. It integrates LLM and VLM guidance to enrich prompts and adapt to HOI tasks effectively. By learning from related seen classes, EZ-HOI overcomes the limitation of lacking labels for unseen classes, enhancing its performance on them. The framework achieves state-of-the-art results with significantly fewer trainable parameters than existing methods, showcasing its efficiency and effectiveness in zero-shot HOI detection.

**Strengths:**

- The paper is well-written.
- The Unseen Text Prompt Learning is well-designed.
- Overall, the zero-shot results with less trainable parameters are good.

**Weaknesses:**

- This paper primarily addresses the improvement of zero-shot HOI detection benchmark performance by training with the pre-definition of all HOI classes. This creates an unfair comparison with previous work.
- In light of the above, the reviewer believes that the authors should provide experimental results where the pre-definition of all HOI classes is unknown during training, allowing for a fairer comparison with previous work.

**Questions:**

N/A

**Limitations:**

Yes

---

> ### Author Rebuttal · Authors · 2024-08-07
>
> We appreciate the positive feedback. The following is our response for your questions. Note that references maintain their original numbering from the main paper, with new references denoted by letters and listed at the end.
>
> ## 1. Pre-definition of all HOI classes
>
> Regarding the comparison with previous work, we follow the common practices in the zero-shot HOI detection setting, as introduced in the following. Zero-shot HOI setting involves predicting unseen HOI classes, where unseen class names are typically used in training [13,15,14,45,38]. In particular, VCL, FCL and ATL [13,15,14] "compose novel HOI samples" during training with the unseen (novel) HOI class names. EoID [45] distills CLIP "with predefined HOI prompts" including both seen and unseen class names. HOICLIP [38] introduces "verb class representation" during training, including both seen and unseen classes.
>
> Beyond the zero-shot HOI setting, there are HOI unknown concept discovery [a] and open-vocabulary HOI detection [b], where unseen class names cannot be used in training. The open-vocabulary setting differs from HOI unknown concept discovery, with a much wider range of unseen HOI classes during testing. We will explore these directions in our future work.
>
>
> ## 2. Experiments without pre-definition of all HOI classes
>
> We appreciate the suggestion to conduct experiments without pre-definition of all HOI classes. As explained in the above question, our work follows the established zero-shot HOI detection protocols, where the use of unseen class names is a common practice, as illustrated by the existing HOI detection methods [13,15,14,45,38]. Nonetheless, we recognize the value of exploring alternative settings where unseen class names are not pre-defined, which could further demonstrate the effectiveness of our approach. While this falls outside the scope of the current submission, we are willing to include this experiment in the final version of our paper, to comprehensively validate our method across varying settings.
>
>
> ## References:
>
> [a] Discovering Human-Object Interaction Concepts via Self-Compositional Learning, ECCV22
>
> [b] Exploring the Potential of Large Foundation Models for Open-Vocabulary HOI Detection, CVPR24

---

> > ### Comment · Area_Chair_VWco · 2024-08-12
> > **Please read through rebuttal, and see if the rebuttal has addressed your concerns or you have further comments.**
> >
> > Hi dear reviewer,
> >
> > Please read through rebuttal, and see if the rebuttal has addressed your concerns or you have further comments.
> >
> > Thanks,
> >
> > AC

---

> > > ### Comment · Reviewer_kLJ1 · 2024-08-13
> > >
> > > Thanks for the authors' response. The rebuttal has addressed most of my concerns. I keep my positive score.

---

> > > > ### Author Response · Authors · 2024-08-13
> > > >
> > > > Thank you for your valuable support. We're pleased that our rebuttal has addressed most of the concerns. We will incorporate the clarification of zero-shot HOI setting configurations in our updated paper.
> > > >
> > > > Regarding the previous feedback from the reviewer on the experiment without pre-defining all HOI classes during training, we believe it would be helpful to expand on this point. Although this experiment, involving unseen class names during training, is beyond the scope of our submission, we would still like to provide the results below to demonstrate our method's effectiveness beyond the zero-shot setting.
> > > >
> > > > As shown in the table below, without unseen class names in training, our model still outperforms the SOTA method UniHOI [3], especially for unseen performances. This experiment highlights the potential applicability of our method beyond the zero-shot HOI setting.
> > > >
> > > >
> > > > | Method |Setting| Full | Unseen | Seen |
> > > > | :-----| :----: | :----: |:----: |:----: |
> > > > | UniHOI|UV |34.68|26.05|36.78|
> > > > |Ours w/o unseen class names |UV|__36.12__ |__28.24__| __37.41__ |
> > > > | UniHOI|RF-UC |32.27|28.68|33.16|
> > > > |Ours w/o unseen class names |RF-UC|__36.96__ |__32.10__ |__38.17__|
> > > > | UniHOI|NF_UC|31.79|28.45|32.63|
> > > > |Ours w/o unseen class names |NF_UC|__34.26__| __34.80__ | __34.13__|

---

### Official Review · Reviewer_RCVD · 2024-07-12

**Soundness:** 2
**Presentation:** 2
**Contribution:** 1
**Rating:** 4
**Confidence:** 5

**Summary:**

This paper investigates the problem of human-object interaction (HOI) detection. This paper introduces EZ-HOI, a method for efficient zero-shot HOI detection in an open-world setting. EZ-HOI also explores the use of LLM and VLM guidance for learnable prompts to enhance prompt knowledge and aid in adapting to HOI tasks. To better adapt to unseen classes, EZ-HOI establishes relationships between unseen and seen classes.

**Strengths:**

- The state-of-the-art experimental results in the zero-shot setting. EZ-HOI demonstrates good performance in zero-shot scenarios. More importantly, by leveraging prompt tuning, the number of trainable parameters in EZ-HOI is significantly smaller compared to other methodologies.
- The code in the supplementary material is provided to ensure the reproducibility of the study.

**Weaknesses:**

- The overall method is straightforward but lacks sufficient novelty. Compared to UniHOI, which also claims to utilize Spatial Prompt Learning, the main difference in EZ-HOI lies in its use of more traditional prompt learning methods (like VPT-DEEP) and its additional modeling of relationships between unseen and seen classes when handling unseen categories. Since many previous works have already demonstrated that prompt tuning (ViT-adapter/CLIP-adapter) can achieve even higher performance than full fine-tuning, the claims of EZ-HOI seem somewhat weak. Additionally, modeling the relationship between unseen and seen classes as a technical contribution also appears somewhat limited.
- The fully-supervised setting results are provided in Appendix Table 6, but the performance is worse than Uni-HOI. Firstly, this is a very important experiment and should be included in the main text. Secondly, it appears that under full supervision, EZ-HOI performs worse than Uni-HOI, which contradicts the claims in the main text. Have the authors provided an in-depth analysis of the reasons? If the performance difference is due to parameter count, what would the performance be with full fine-tuning?
- The performance improvement seems to come from the baseline rather than the proposed modules. As shown in the ablation study (Table 4), the baseline of EZ-HOI achieves 37.44 on seen categories, which is already state-of-the-art (SOTA). I wonder what EZ-HOI’s baseline is and why it can achieve such strong performance on seen categories. Additionally, considering that the UTPL module increases unseen category performance by two points, and EZ-HOI shows a two-point increase compared to Uni-HOI, does this mean that EZ-HOI’s performance gain over Uni-HOI is primarily due to the UTPL module?

**Questions:**

What is the fully fine-tuned performance? What is the baseline?

**Limitations:**

This submission discusses limitations.

---

> ### Author Rebuttal · Authors · 2024-08-07
>
> We appreciate the insightful feedback. Here is our response. Note all references are from the main paper’s citations.
>
> ## Weakness-1. Our novelty and technical contribution
> __(1) Novelty__
>
> We would like to clarify that our innovation lies in proposing a novel framework, rather than a new approach to prompt learning.  Existing methods such as UniHOI [3], HOICLIP [38] and GEN-VLKT [28] align HOI model's visual features to text features from a frozen VLM. However, aligning with VLM features requires training transformer-based models, which is computationally expensive.
>
> Unlike existing methods that utilize frozen VLM text features, we fine-tune both the visual and text encoders of the VLM with guidance from an LLM. This allows us to adapt both visual and text features to our HOI setting, rather than only adapting visual features. Consequently, our method, EZ-HOI, improves the alignment between visual and text representations, achieving SOTA performance in zero-shot HOI settings with significantly fewer trainable parameters.
>
> A comparison between our method and UniHOI is shown in the table below. Like most existing HOI detection methods [38,28,36,45], we use CLIP as the VLM. UniHOI [3], on the other hand, uses BLIP2, a more powerful VLM than CLIP. As shown in [25], the performance of BLIP2 is significantly superior to CLIP in zero-shot image-text retrieval tasks.
>
> Despite using CLIP, our method still achieves comparable performance to UniHOI under various zero-shot settings, which demonstrates the effectiveness of our novel framework, EZ-HOI.
>
> | Method |VLM|Setting| Full | Unseen | Seen |
> | :-----| :----: | :----: |:----: |:----: |:----: |
> | UniHOI|BLIP2|UV |34.68|26.05|36.78|
> |Ours |CLIP|UV|__36.84__|__28.82__|__38.15__|
> | UniHOI|BLIP2|RF-UC |32.27|28.68|33.16|
> |Ours |CLIP|RF-UC|__36.73__|__34.24__|__37.35__|
> | UniHOI|BLIP2|NF_UC|31.79|28.45|32.63|
> |Ours |CLIP|NF_UC|__34.84__|__36.33__|__34.47__|
>
> __(2) Technical contributions__
>
> Regarding modeling the relationship between unseen and seen classes, our novelty lies in the UTPL module. Our UTPL enhances unseen prompt learning by leveraging related seen classes with the help of LLM-generated descriptions, which to our knowledge is a novel idea. This is in contrast to existing methods, which either use LLM descriptions without seen class context [3] or rely on seen class information alone [13,15].
>
> Our technical contribution is centered around mitigating the overfitting problem when using prompt learning to adapt a VLM for zero-shot HOI detection. While existing prompt learning typically suffers from the overfitting problem to seen classes [56,6,8,19], our method achieves 11.6 mAP improvement on unseen HOI classes compared the prompt learning baseline (MaPLe [19]) as shown in Fig. 1(d).
>
> ## Weakness-2. Performance under fully-supervised setting
>
> In our Appendix, Table 6 presents the quantitative comparison under fully-supervised settings. To ensure a fair comparison, we used CLIP as our VLM, the same as most existing methods [38,28,36,45]. As mentioned earlier in this rebuttal, UniHOI employs BLIP2, which is a superior VLM compared to CLIP.
>
> To provide a fair comparison, we show the performances of UniHOI and our method using the same VLM (CLIP). As observed, our method outperforms UniHOI when both use CLIP.
>
> The results for UniHOI using CLIP were obtained from UniHOI's OpenReview rebuttal (<https://openreview.net/forum?id=pQvAL40Cdj&noteId=kI6HJnJ1KB>).
>
> | Method | Full | Rare | Non-rare |
> | :-----| :----: | :----: |:----: |
> | $UniHOI_l$ |36.84|35.71|37.05|
> |Ours |38.61|37.70|38.89|
>
> We will follow the suggestion to move Table 6 of the appendix to the main text, and include our rebuttal discussion to the paper upon acceptance.
>
> ## Weakness-3. Baseline method and performance gain
>
> __(1) Baseline and the strong seen performance__
>
> Our baseline is MaPLe [19], not the first row of Table 4 in our main text. The first row (Table 4) is part of our developed method, focusing on improving seen classes, which employs our intra-HOI fusion and visual adapter. Both modules are mentioned in Section 3.3: Intra-HOI is introduced in Appendix 6.4, and visual adapter is based on [23].
>
> To clarify the baseline performance, we provide an extended ablation study table below. The first row shows the baseline method, MaPLe [19]. Our intra-HOI fusion module improves the seen HOI performance by 7.41 mAP, as shown in the second row. The third row is the result of adding the visual adapter, which is the first row of Table 4 in our main text. We will follow the suggestion to update the ablation studies in Table 4 of the main paper and include this discussion.
>
>
> | Method |Intra-HOI fusion|visual adapter [23]| Full | Unseen | Seen |
> | :-----| :----: | :----: |:----: |:----: |:----: |
> | MaPLe |No|No| 26.26|17.19|27.73|
> |MaPLe|Yes|No|33.52|23.54|35.14|
> |MaPLe| Yes|Yes|35.40|22.91|37.44|
> |Ours (full model)|Yes|Yes|38.61|37.70|38.89|
>
> __(2) Performance gain from each designed modules__
>
> Since our baseline is MaPLe [19], the performance gain over UniHOI is not solely due to the UTPL module but results from all the modules we designed (including LLM guidance, VLM guidance, and intra- and inter-HOI fusion). Each module's contribution is shown in the table above and in Table 4 of the main paper, illustrating how each module incrementally enhances the final performance. While it is true that UTPL contributes 2.42 mAP to unseen class performance, other modules, such as our LLM guidance, also significantly improve unseen performance by 1.52 mAP, and VLM guidance increases unseen performance by 1.33 mAP.
>
> ## Question-1. Fully fine-tuned method and baseline
> As discussed in Question 3, our baseline is MaPLe [19]. Our framework is based on prompt tuning to enable a VLM to adapt to HOI tasks, so we do not use full fine-tuning in our method. We acknowledge that full fine-tuning might be beneficial, and we plan to explore this in our future work.

---

> > ### Comment · Area_Chair_VWco · 2024-08-12
> > **Please read through rebuttal, and see if the rebuttal has addressed your concerns or you have further comments.**
> >
> > Hi dear reviewer,
> >
> > Please read through rebuttal, and see if the rebuttal has addressed your concerns or you have further comments.
> >
> > Thanks,
> >
> > AC

---

> > > ### Author Response · Authors · 2024-08-14
> > >
> > > Dear reviewer, with the deadline for our discussion approaching, we kindly ask if you have any remaining concerns or feedback. Your insights are highly valuable to us, and we would be grateful for any further comments or questions you may have.

---

### Official Review · Reviewer_1u4N · 2024-07-27

**Soundness:** 3
**Presentation:** 3
**Contribution:** 3
**Rating:** 6
**Confidence:** 5

**Summary:**

The paper proposes a novel prompt learning framework for zero-shot Human-Object Interaction (HOI) detection, which enhances the generalizability of Vision-Language Models (VLMs) by interacting with Large Language Models (LLMs) to obtain descriptions.

**Strengths:**

1. The proposed method appears logical and achieves general improvements in performance.
2. Utilizing LLMs to generate descriptions is a promising direction.
3. The paper is well-written, and the framework diagram is clear.

**Weaknesses:**

1. The prompt design seems to be a hybrid approach, combining visual prompt learning [1, 3] with text prompts, such as those used in COCOOP, without a specific focus on task-specific configurations. It is difficult to justify the necessity of this work for the field.

2. Despite the claim of improved performance on novel classes, the method performs worse on unseen classes compared to CLIP4HOI, while it shows improved performance on seen classes. Given that novel class performance is crucial, this result seems not consistent with this task.

3. W_down presents in both Equation 3 and Equation 5 is puzzling. Equation 5 attempts to integrate descriptions for unknown categories with similar known categories and LLM-generated details. More details on the training process would be helpful, as it appears that data training might be lacking since unknown categories are not included in the training.

4. There is substantial related work on using LLMs for new class descriptions in zero-shot scenarios, such as CuPL [2] for classification and DVDet [3] for detection, which use attribute decomposition to aid in categorization. DVDet also generates more discriminative descriptors by distinguishing confusing categories. The paper lacks a discussion on such related LLM-based works. It would be interesting to see if using action descriptors might also yield some benefits.

5. The paper could benefit from providing more visualization results to demonstrate the effectiveness of both general and discriminative descriptions generated by the method.

[1] Visual Prompt Tuning

[2] VISUAL CLASSIFICATION VIA DESCRIPTION FROM LARGE LANGUAGE MODELS

[3] LLMS MEET VLMS: BOOST OPEN-VOCABULARY OBJECT DETECTION WITH FINE-GRAINED DESCRIPTORS

**Questions:**

Please see the weakness.

**Limitations:**

Yes.

---

> ### Author Rebuttal · Authors · 2024-08-07
>
> Thank you for your detailed, helpful feedback. We address each of your concerns in the following.
>
> ## 1. Task-specific configurations
>
> Our method is specifically designed for HOI settings with the following task-specific configurations.
>
> First, our method is configured for HOI detection by extracting visual features for each human-object pair in an image, different from using visual features from an entire image like general prompt learning methods [19,56,6,8]. We extract HOI visual features from each human-object pair to compare with text features, which is introduced in Appendix Section 6.4.
>
> Second, the UTPL module is designed for HOI settings to recognize interactions with strong connections and subtle differences (i.e., "straddle bike" and "ride bike"). If "ride bike" is unseen, UTPL leverages related seen HOIs like "straddle bike" to help with recognizing "ride bike".
>
> Third, our proposed LLM-generated description is another task-specific configuration. In particular, the LLM-generated descriptions provide the nuances between unseen and related seen HOI classes, which further enhance the understanding of the unseen class.
>
> ## 2. Comparison with CLIP4HOI
>
> We acknowledge our method’s slightly lower performance when compared to CLIP4HOI under the unseen verb setting, where our method's unseen performance is 0.92 mAP lower. While it is true that there is a minor drop in accuracy, it is important to consider the significant reduction in trainable parameters that our method achieves—87.9% fewer than CLIP4HOI. Moreover, under the non-rare first unseen composition setting, we outperform CLIP4HOI by 2.22 mAP for unseen HOI classes (Table 2 second column from right to left, of the main paper).
>
> This substantial decrease in model complexity not only lowers the computational cost but also makes it more accessible to researchers in the field with limited resources. Achieving a balanced trade-off between performance and efficiency is essential, which aligns with the growing need for more efficient models in the field. Our method provides a practical solution by offering competitive performance while drastically reducing the resource requirements. Thus, the minor compromise in accuracy is counterbalanced by significant gains in model efficiency and accessibility.
>
>
> ## 3. Clarification and training details for Eqs. (3),(5)
>
> $W_{down}$ in Eqs. (3) and (5) refer to the down projection layers, but they are not the same projection layers with different weights. We provide more training details for Eq. (5) in the following.
>
> Although there is no annotated image for the unseen/unknown categories in training, we have two ways to optimize Eq. (5). First, we design a class-relation loss (Eq. (15) in the Appendix) to keep the relationship between seen and unseen classes, measured by cosine similarity between text features. This way, unseen prompts can also be refined based on their relation to seen classes. Second, while there are no annotated images for unseen/unknown classes, the annotated training data serves as negative samples. If the prediction score for an unseen class is too high, the model is penalized (Eq. (16) in the Appendix).
>
> Additionally, in Eq. (5), each unseen learnable prompt is linked to a similar seen learnable prompt. If the seen prompts are optimized after each training step, the updated seen prompts will be used to refine the corresponding unseen prompts in UTPL. We will include the above discussion in the updated paper.
>
>
> ## 4. Benefits from action descriptors
>
> We thank the reviewers for pointing out the related LLM-based works such as action descriptors, which are potentially useful for HOI detection. We conducted the following preliminary experiment given the limited time in this rebuttal.
>
> Descriptors mentioned in CuPL[a] and DVDet [b] refer to attribute decomposition to benefit category recognition. We leverage the attribute decomposition idea from DVDet and tailor it to our method. We follow DVDet [b] to generate action descriptors for each class and integrate them into HOI class text features. This process enhances the detail and distinctiveness of the HOI class representations. Descriptors with low cosine similarity to the HOI class text features are discarded to avoid noisy information.
>
> The following table presents our preliminary result under the unseen verb setting, showing improvement with our simple and direct adoption of the action descriptors. These results demonstrate that LLM-generated descriptions, such as action descriptors, are potentially useful for HOI detection. We will include a discussion of the related LLM-based works in our updated paper.
>
> | Methods |Full | Seen | Unseen |
> | :-----| :----: | :----: |:----: |
> |Ours|32.32|25.10|33.49|
> |Ours + action descriptor| 32.63 | 25.14 | 33.85|
>
>
> ## 5. Visualization results
>
> We acknowledge the reviewer’s request for additional visualizations. While we have made every effort to understand and address this, with all due respect, we are still unsure what visualization results we need to add. Based on our understanding, in this rebuttal, we provide the qualitative results to illustrate the impact of LLM guidance (using general description) and UTPL (using discriminative description) shown in Fig. 1 of our attached PDF.
>
> The LLM guidance provides detailed class information, improving the performance over the baseline. Distinctive descriptions help to distinguish unseen classes from related seen HOIs, enhancing unseen performance and challenging case predictions.
>
>
> ## References
>
> [a] Visual classification via description from large language models, ICLR23
>
> [b] LLMs meet VLMs: Boost open vocabulary object detection with fine-grained descriptors, ICLR24

---

> > ### Comment · Area_Chair_VWco · 2024-08-12
> > **Please read through rebuttal, and see if the rebuttal has addressed your concerns or you have further comments.**
> >
> > Hi dear reviewer,
> >
> > Please read through rebuttal, and see if the rebuttal has addressed your concerns or you have further comments.
> >
> > Thanks,
> >
> > AC

---

> > ### Comment · Reviewer_1u4N · 2024-08-13
> >
> > The author addressed most of my concerns, and the visualized results are also reliable, so I have decided to increase my score.

---

> > > ### Author Response · Authors · 2024-08-13
> > >
> > > Thank you for the positive comments and for raising the score. We're pleased that our rebuttal addressed most of your concerns. Based on your suggestions, we will include the visualized results and discussions in our paper.

---

### Official Review · Reviewer_kbXP · 2024-07-30

**Soundness:** 3
**Presentation:** 2
**Contribution:** 3
**Rating:** 6
**Confidence:** 3

**Summary:**

The paper studies the challenges behind successful adaption of pre-trained vision language models (VLMs), i.e., CLIP, to the problem of zero-shot Human Object Interaction (HOI) detection. Specifically, during finetuning, VLMs overfit to seen HOI classes observed during the training on HOI training data, preventing successful transfer on the unseen HOI classes. To overcome the aforementioned problem, authors propose prompt tuning mechanism coupled with guidance from a large language model (LLM). In particular, LLM is used to generate (i) descriptions of each HOI class; (ii) explain the difference between each unseen and the corresponding closest seen class to facilitate transfer on unseen classes. The prompt tuning technique itself learns a set of prompts that are shared between vision and text encoders. Authors employ attention mechanism to produce modality specific prompts via attending on class descriptions for text modality and via attending on the image embedding for visual modality correspondingly. Authors evaluate the proposed approach on the standard HOI benchmark and compare it to the recent baselines, showing improvements in terms of the parameter efficiency and the performance on unseen classes.

**Strengths:**

* The problem of enabling generalization to unseen classes during VLMs adaptation is important, even outside HOI detection field
* To my knowledge, learning shared prompts and producing modality specific prompts via attention mechanism is novel
* The evaluation is thorough, both comparing to HOI specific prompt tuning baselines and to general purpose prompt tuning approaches such as MaPLe.

**Weaknesses:**

My main concern is that the current narration of the proposed methodology lacks intuition and structure. For example,

**Structure**
* Section 3 starts with mentioning that the method will learn prompts per layer, and the subsequent section also use notation considering N layers, but the first time the reader encounters interaction with the layers is Section 3.3. It makes the paper hard to follow.

**Intuition**
* The main idea of the approach builds on repeatedly applying MHCA for different purposes, however these equations just stated on the high-level in Eq. (2); (3); (5) and without any intuition behind the design.

* Similarly to the above, Eq. (6) and Eq. (8) state how these learnable prompts are used in the corresponding encoder layers, however the narration is on the high level and does not elucidate the particular idea behind the equations.

**Questions:**

* How $N=9$ was chosen? Can you provide ablations for different number of $N$?
* Similarly to the question above, Eq. (6) and (8) suggest that the learnable prompts are inserted in the first $N$ layers of textual and visual encoders, correspondingly. What is the motivation behind this design choice? I, in general, would assume that earlier layers already consumed broad knowledge during CLIP pre-training and task specific adaptation is required for penultimate layers. Can you please provide ablations on different positioning of the learnable prompts?
* Currently, the paper narration mainly focuses on tackling generalization to unseen classes and, indeed, experimentally confirms the improvements offered by the proposed approach. The main component causing such improvements is seem to be UTPL that employs guidance from LLM. However, according to Table 4, other components also bring substantial improvements. Can you please provide the intuition behind other components' role and how they can bring these improvements?

**Limitations:**

Authors discuss the limitations of the current zero-shot HOI detection setting. In particular, that to truly enable zero-shot HOI detection, one must exclude the assumption of having in advance a set of unseen HOI classes.

---

> ### Author Rebuttal · Authors · 2024-08-07
>
> We appreciate the positive feedback and detailed reviews. The following is our response for your questions. Note that we use the same reference number for related papers as the main paper.
>
> ## Weakness-1. Structure of the paper
> We agree with the suggestion to introduce the "encoder layer" in Section 3.3 rather than at the beginning of Section 3, and will revise our paper.
>
> ## Weakness-2. Design intuition for Eqs. (2), (3), (5)
> Here we provide detailed design intuition for Eq. (2) about LLM guidance, Eq. (3) about VLM guidance and Eq. (5) about UTPL.
>
> In Eq. (2), the text learnable prompts $\mathcal H_T$ are integrated with text embeddings $\mathcal F_{txt}$ from LLM-generated class description. Through MHCA, $\mathcal H_T$ only aggregate useful information from $\mathcal F_{txt}$ with learnable attention. This is important because the information provided by LLM may not be equally crucial for our task.
>
> Moreover, specific design is introduced and used in Eqs. (3) and (5).
> To keep trainable parameters small, we apply a down-projection layer $W_{down}$ before MHCA to reduce the feature dimension, and an up-projection layer $W_{up}$ afterward. In addition, initially Eq. (2) output equals to input $\mathcal H_T$, by initializing $W_{up}$ to 0, which stabilizes training by gradually fine-tuning $\mathcal H_T$.
>
> In Eq. (3), visual features $f_{vis}^{\mathcal I}$ from the VLM are integrated with visual learnable prompts $\mathcal H_V$ by MHCA. Since the frozen VLM visual encoder can extract features for unseen HOIs, $\mathcal H_V$ aggregates information from these visual features, improving performance on unseen HOIs.
>
> In Eq. (5), the unseen learnable prompts $\hat{h_{T_u}}$ are combined with disparity information $f_{ txt_u}^{\rm disp}$, related seen class prompts $\hat{h_{T_s}}$, and the unseen class prompts themselves $\hat{h_{T_u}}$, in MHCA. The disparity information $f_{ txt_u}^{\rm disp}$  provides distinctive attributes $\hat{h_{T_u}}$. The related seen class prompts $\hat{h_{T_s}}$ enhance $\hat{h_{T_u}}$ by transferring shared features to unseen classes. The unseen class prompts $\hat{h_{T_u}}$ retain self-information and are emphasized through MHCA's processing.
>
>
> ## Weakness-3. Design intuition for Eqs. (6) and (8)
> Here we provide detailed design intuition for Eq. (6) for deep text prompt learning and Eq. (8) for deep visual prompt learning.
>
> In deep text prompt learning, we put the learnable text prompts $\tilde{h_T^i}$ at the end of the original text prompts $W_i$.
> $W_1$, in the first layer, is generated from "a photo of a person \<acting> a/an \<object>".
> Since we design $\tilde{h_T^i}$ to be class-specific with HOI class information, its semantic information is very different from "a photo of", related to the beginning position of $W_1$.
> Therefore, the end position of $W_1$ is better suited for $\tilde{h_T^i}$, where the semantic information is more connected to $W_1$.
>
>
> Moreover, after N layers, we stop introducing new learnable prompts.
> Basically, when introducing new learnable prompts to each layer, we observe the VLM deals with seen classes better with decreased unseen performance.
> Moreover, inserting learnable prompts into deeper layers of VLM, makes the performance sensitive, because the feature space is mature in those layers [19].
>
> Similar to deep text prompt learning, deep visual prompt learning also only introduces new learnable prompts $\hat{h_V^i}$ until layer N.
> The position of $\hat{h_V^i}$ does not significantly influence the outcome, as original frozen visual prompts $E_i$ correspond to different regions of the input image, but $\hat{h_V^i}$ contains global image information, equally enhancing $E_i$.
>
> ## Question-1. Ablation study for hyperparameter $N$
> The following table shows the ablation study for the hyperparameter $N$, where $N$ means we introduce learnable prompts from the first layer until layer $N$.
>
> | N | Full | Seen | Unseen |
> | :-----| :----: | :----: |:----: |
> | 4 |32.60|33.99|24.10|
> |9|32.32|33.49|__25.10__|
> |12|32.76|34.19|23.98|
>
> We use N=9 in our main paper because it shows the best unseen performance.
>
> ## Question-2. Ablation study for different positions of learnable prompts
> The following table shows the ablation study for the different positions of learnable prompts. We find that the position of learnable prompts does not affect the outcome too much. Thus, we follow [19] and fine-tune layers 1-12, where fine-tuning layers 1-9 shows slightly better unseen performance.
>
> | Position | Full | Seen | Unseen |
> | :-----| :----: | :----: |:----: |
> | 1-9 |32.32|33.49|__25.10__|
> |3-11| 32.24| 33.45 | 24.83 |
> |4-12|32.40 |33.64| 24.82 |
>
> ## Question-3. Intuition for components tackling generalization, other than UTPL
> We introduce components including LLM guidance, VLM guidance and inter-HOI fusion, other than the UTPL module, which can enhance generalization ability to unseen classes. We discuss the design intuition one by one.
>
> The LLM guidance, mentioned in Section 3.1 "Text Prompt Design" of the main paper, integrates learnable text prompts with detailed HOI class descriptions. This approach enhances the model's understanding of unseen classes, which lack training data, by providing detailed information rather than simple class names.
>
> The VLM guidance, detailed in Section 3.1 "Visual Prompt Design," combines learnable visual prompts with image features from the frozen VLM visual encoder. Since the VLM includes unseen HOI information, the frozen encoder can extract unseen HOI representations during testing. The learnable visual prompts then aggregate these representations, enhancing unseen class prediction.
>
> Inter-HOI fusion, as mentioned in Appendix Section 6.4, refines HOI visual features by considering surrounding HOI features in the image. For instance, to detect "cut a cake," the model uses the surrounding visual context like "cut with a knife," making recognition easier.

---

> > ### Comment · Reviewer_kbXP · 2024-08-09
> > **Acknowledgement**
> >
> > I thank the authors for the provided clarifications. After reading the other reviewers responses and the rebuttal I will maintain my positive score. I strongly suggest authors to include the provided clarifications in the future revision of the manuscript. Given that, partially, the proposed approach employs well-established techniques in the prompt tuning community, I also believe that the paper will greatly benefit from expanding discussion and positioning with respect to these existing techniques.

---

> > > ### Author Response · Authors · 2024-08-11
> > >
> > > Thank you for the positive feedback and valuable suggestions. We're glad that our rebuttal has addressed your concerns. We will include the provided clarifications in our paper, and also expand the discussion on existing prompt-tuning techniques in our paper as suggested. If there are any needs for further information or clarification, please let us know.

---

### Official Review · Reviewer_UfLd · 2024-07-31

**Soundness:** 3
**Presentation:** 3
**Contribution:** 2
**Rating:** 5
**Confidence:** 3

**Summary:**

This paper tackles zero-shot HOI detection via prompt tuning. To address the challenge posed by the absence of novel classes, the authors first incorporate LLM and VLM guidance to enrich learnable prompt tokens. Further, the authors utilize LLM to provide nuanced differentiation between unseen classes and their related seen classes, termed Unseen Text Prompt Learning (UTPL), to alleviate overfitting to seen classes. In experiment, the proposed method achieves state-of-the-art performance under major zero-shot HOI detection settings while costing fewer training resources.

**Strengths:**

1. This paper tackles an important issue in zero-shot / open-vocabulary learning, i.e., overfitting to seen classes.
2. The proposed method effectively explores guidance from large foundation models, including VLM and LLM.
3. The authors achieve state-of-the-art performance with significantly fewer training resources.

**Weaknesses:**

1. It seems the authors assume the prior knowledge of unseen class names when training the UTPL module, which conflicts with the zero-shot setting.

2. In Table 1, the proposed method underperforms CLIP4HOI (ResNet50+ViT-B) in unseen mAP although it is claimed to tackle the overfitting issue especially.

**Questions:**

1. In Figure 2, are the VLM visual and visual encoder the same thing (i.e., CLIP)?

2. Why did the authors use LLaVA for text description instead of choosing pure language models?

---

> ### Author Rebuttal · Authors · 2024-08-07
>
> We appreciate the positive feedback and insightful comments. The following is a detailed response to each of your questions. Note that references maintain their original numbering from the main paper, with new references denoted by letters and listed at the end.
>
>
> ## Weakness-1. Use of unseen class names
>
> We follow the common practices in the zero-shot HOI detection setting. Zero-shot HOI setting involves predicting unseen HOI classes, where unseen class names are typically used in training [13,15,14,45,38]. In particular, VCL, FCL and ATL [13,15,14] "compose novel HOI samples" during training with the unseen (novel) HOI class names. EoID [45] distills CLIP "with predefined HOI prompts" including both seen and unseen class names. HOICLIP [38] introduces "verb class representation" during training, including both seen and unseen classes.
>
> Apart from using unseen class names, the prior knowledge of unseen class names is also used in existing work. UniHOI [3] utilizes LLM-generated descriptions, the prior knowledge of the unseen class names, to recognize unseen classes better with "rich and detailed representation". Thus, our method follows the common practices in the zero-shot HOI detection setting for conducting experiments.
>
> Beyond zero-shot HOI settings, there are HOI unknown concept discovery [a] and open-vocabulary HOI detection [b], where unseen class names cannot be used in training.
> The open-vocabulary setting differs from HOI unknown concept discovery, with a much wider range of unseen HOI classes during testing.
> We will explore these directions in our future work.
>
> ## Weakness-2. Comparison with CLIP4HOI
> We acknowledge our method’s slightly lower performance when compared to CLIP4HOI under the unseen verb setting, where our method's unseen performance is 0.92 mAP lower. While it is true that there is a minor drop in accuracy, it is important to consider the significant reduction in trainable parameters that our method achieves—87.9% fewer than CLIP4HOI. Moreover, under the non-rare first unseen composition setting, we outperform CLIP4HOI by 2.22 mAP for unseen HOI classes (Table 2 second column from right to left, of the main paper).
>
> This substantial decrease in model complexity not only lowers the computational cost but also makes it more accessible to researchers in the field with limited resources. Achieving a balanced trade-off between performance and efficiency is essential, which aligns with the growing need for more efficient models in the field. Our method provides a practical solution by offering competitive performance while drastically reducing the resource requirements. Thus, the minor compromise in accuracy is counterbalanced by significant gains in model efficiency and accessibility.
>
> ## Question-1. Figure 2 visual encoder
> In Figure 2, the "VLM visual" is different from the "visual encoder". The "VLM visual" is the frozen CLIP visual encoder, while the "visual encoder" shown in Figure 2 is fine-tuned during training and is based on our design, including visual learnable prompts $\hat{\mathcal{H}}_V$ and the HOI feature fusion module.
>
> ## Question-2. Language model selection
>
> Pure language models can also be applied to our methods for text description generation. We leverage LLaVA because it demonstrates quite strong reasoning results with GPT-4 level capabilities [32].
> As shown in the following, outputs from LLaVA and Chatgpt 3.5 are both informative and reasonable, benefiting the model for learning detailed representation.
>
>
> LLaVA output: _"Swinging a baseball bat" describes a person using a baseball bat to hit a ball. This action typically involves the person holding the bat with both hands, standing in a stance with their feet shoulder-width apart, and using their body rotation to contact the ball._
>
> Chatgpt3.5 output: _"Swing a baseball bat" involves standing with feet apart, gripping the bat, and stepping forward as the pitch approaches. The batter rotates the torso, bringing the bat through the hitting zone to hit the ball._
>
>
> ## References:
>
> [a] Discovering Human-Object Interaction Concepts via Self-Compositional Learning, ECCV22
>
> [b] Exploring the Potential of Large Foundation Models for Open-Vocabulary HOI Detection, CVPR24

---

> > ### Comment · Reviewer_UfLd · 2024-08-10
> >
> > Thanks for the authors' response. My major concerns about the zero-shot setting have been resolved.

---

> > > ### Author Response · Authors · 2024-08-11
> > >
> > > Thank you for your positive feedback. We're pleased that our rebuttal has addressed your concerns.  We will include the detailed discussion for zero-shot HOI setting configuration in our paper.

---

### Author Rebuttal · Authors · 2024-08-07

We thank all reviewers for their thoughtful and constructive feedback. We appreciate that the reviewers found our work to be "innovative" (Reviewer f78r), "tackling an important issue in zero-shot learning and generalization to unseen classes during VLMs adaptation" (Reviewer UfLd, kbXP), and "well-written" (Reviewer 1u4N, kLJ1, f78r).

Regarding the major modules in our method, we thank the reviewers to recognize that "the proposed learnable prompts are novel" (Reviewer kbXP, f78r), and "well-designed Unseen Text Prompt Learning (UTPL)" (Reviewer kLJ1, f78r).

In terms of evaluation and quantitative comparison, we appreciate the reviewers to point out that "the proposed method achieves state-of-the-art performance with significantly smaller trainable parameters" (Reviewer UfLd, RCVD, and kLJ1), and "the evaluation is thorough" (Reviewer kbXP).

---

> ### Comment · Area_Chair_VWco · 2024-08-09
> **kicking off reviewer-author discussion**
>
> Thanks the authors for the rebuttal, and reviewers' for the comments. We are in the period of author-reviewer discussion.
> - Reviewers, please read through all the reviews and rebuttal and see if authors' responses have addressed your and others' important questions.
> - Authors, please follow up with any reviewers' further comments.
>
> Cheers,
>
> AC

---

### Decision · Program_Chairs · 2024-09-25

**Decision:**

Accept (poster)

**Comment:**

The paper was evaluated by six reviewers. Authors provided strong rebuttal and most reviewers engaged in the author-reviewer discussion. As a result, five out of the six reviewers provided positive ratings, and the sixth reviewer who provided negative review did not respond to authors rebuttal. After reading the reviews and rebuttal, the AC agrees to accept this paper. Authors are encouraged to polish the paper further by considering the questions, suggestions and comments of the reviewers.